# Fearless Stochasticity in Expectation Propagation

**Jonathan So**
University of Cambridge
js2488@cam.ac.uk

**Richard E. Turner**
University of Cambridge
The Alan Turing Institute

## Abstract

Expectation propagation (EP) is a family of algorithms for performing approximate inference in probabilistic models. The updates of EP involve the evaluation of moments—expectations of certain functions—which can be estimated from Monte Carlo (MC) samples. However, the updates are not robust to MC noise when performed naively, and various prior works have attempted to address this issue in different ways. In this work, we provide a novel perspective on the moment-matching updates of EP; namely, that they perform natural-gradient-based optimisation of a variational objective. We use this insight to motivate two new EP variants, with updates that are particularly well-suited to MC estimation. They remain stable and are most sample-efficient when estimated with just a single sample. These new variants combine the benefits of their predecessors and address key weaknesses. In particular, they are easier to tune, offer an improved speed-accuracy trade-off, and do not rely on the use of debiasing estimators. We demonstrate their efficacy on a variety of probabilistic inference tasks.

## 1 Introduction

Expectation propagation (EP) [37, 40] is a family of algorithms that is primarily used for performing approximate inference in probabilistic models [9, 10, 12, 13, 16, 18, 20, 19, 21, 22, 23, 24, 28, 29, 30, 32, 35, 36, 38, 39, 41, 44, 47, 48, 50, 52, 53], although it can be used more generally for approximating certain kinds of functions and their integrals [14].

EP involves the evaluation of moments—expectations of certain functions—under distributions that are derived from the model of interest. EP is usually applied to models for which these moments have convenient closed-form expressions or can be accurately estimated using deterministic methods. Moments can also be estimated using Monte Carlo (MC) samples, significantly expanding the set of models EP can be applied to. However, the updates of EP are not robust to MC noise when performed naively, and various prior works have attempted to address this issue in different ways [17, 48, 52].

In this work we provide a novel perspective on the moment-matching updates of EP; namely, that they perform natural-gradient-based optimization of a variational objective (Section 3). We use this insight to motivate two new EP variants, EP-$\eta$ (Section 3.2) and EP-$\mu$ (Section 3.3), with updates that are particularly well-suited to MC estimation, remaining stable and being most sample-efficient when estimated with just a single sample. These new variants combine the benefits of their predecessors and address key weaknesses. In particular, they are easier to tune, offer an improved speed-accuracy trade-off, and do not rely on the use of debiasing estimators. We demonstrate their efficacy on a variety of probabilistic inference tasks (Section 4).

## 2 Background

In this section, we first introduce the problem setting. We then give an overview of EP, followed by a discussion of issues related to sampled estimation of EP updates.

38th Conference on Neural Information Processing Systems (NeurIPS 2024).

Let $\mathcal{F}$ be the tractable, minimal exponential family of distributions (see Appendix A), defined by the statistic function $s(.)$ with respect to base measure $\nu(.)$. Let $\Omega$, $\mathcal{M}$, $A(.)$ denote the natural domain, mean domain and log-partition function of $\mathcal{F}$, respectively, and $A^*(.)$ the convex dual of $A(.)$.

Let $p_0$ be the member of $\mathcal{F}$ with natural parameter $\eta_0$, so that $p_0(z) = \exp(\eta_0^\top s(z) - A(\eta_0))$[1], and assume that a distribution of interest $p^*$, the *target distribution*, has a density of the form

$$p^*(z) \propto p_0(z) \prod_{i=1}^{m} \exp\big(\ell_i(z)\big). \tag{1}$$

In Bayesian inference settings, $p^*$ would be the posterior distribution over parameters $z$ given some observed data $\mathcal{D}$, where $p_0$ may correspond to a prior distribution, and $\{\ell_i(z)\}_i$ to log-likelihood terms given some partition of $\mathcal{D}$.[2] However, we consider the more general setting in which the target distribution is simply a product of factors. Note that we can assume form (1) without loss of generality if we allow $p_0$ to be improper. The inference problem typically amounts to computing quantities derived from the normalised density $p^*(z)$, such as samples, summary statistics, or expectations of given functions. In some special cases, these quantities can be computed exactly, but this is not feasible in general, and approximations must be employed.

## 2.1 Expectation propagation (EP)

Given a target distribution density of form (1), EP aims to find an approximation $p \in \mathcal{F}$ such that

$$p(z) \propto p_0(z) \prod_i \exp\big(\lambda_i^\top s(z)\big) \approx p^*(z). \tag{2}$$

By assumption $\mathcal{F}$ is tractable, and so provided that $(\eta_0 + \sum_i \lambda_i) \in \Omega$, $p$ is a tractable member of $\mathcal{F}$. Each factor $\exp(\lambda_i^\top s(z))$ is known as a *site potential*, and can be roughly interpreted as a $\mathcal{F}$-approximation to the $i$-th *target factor*, $\exp(\ell_i(z))$.[3] $\lambda_i$ is known as the $i$-th *site parameter*.

**Variational problem** EP, and several of its variants [17, 24, 37], can be viewed as solving a variational problem which we now introduce following the exposition of Hasenclever et al. [17].

Let the $i$-th *locally extended family*, denoted $\mathcal{F}_i$, be the exponential family defined by the statistic function $s_i(z) = (s(z), \ell_i(z))$ with respect to base measure $\nu(.)$. Let $\Omega_i$, $\mathcal{M}_i$ and $A_i(.)$ denote the natural domain, mean domain, and log partition function of $\mathcal{F}_i$, respectively. Note that a member of $\mathcal{F}_i$ roughly corresponds to a distribution whose density is the (normalised) product of a member of $\mathcal{F}$ with the $i$-th target factor raised to a power. Unlike $\mathcal{F}$, we do not assume $\mathcal{F}_i$ is minimal.

Fixed points of EP correspond to the solutions of the saddle-point problem

$$\max_{\theta \in \Omega} \min_{\{\lambda_i\}_i} L(\theta, \lambda_1, \ldots, \lambda_m), \text{ where}$$

$$L(\theta, \lambda_1, \ldots, \lambda_m) = A\Big(\eta_0 + \sum_i \lambda_i\Big) + \sum_i \beta_i \big[A_i((\theta - \beta_i^{-1}\lambda_i, \beta_i^{-1})) - A(\theta)\big]. \tag{3}$$

At a solution to (3), the EP approximation is given by (2). The hyperparameters $\{\beta_i\}_i$ control the characteristics of the approximation, and correspond to the power parameters of power EP.

**EP updates** EP [37, 40], power EP [34] and double-loop EP [24], can all be viewed as alternating between some number ($\geq 1$) of *inner updates* to decrease $L$ with respect to $\{\lambda_i\}_i$, with an *outer update* to increase $L$ with respect to $\theta$ [17, 25]. EP is not typically presented in this way, but by doing so we will be able to present a unified algorithm that succinctly illustrates the relationship between the different variants. We show equivalence with the conventional presentation of EP in Appendix B. The inner and outer updates are given by

$$\textbf{Inner update:} \quad \lambda_i \leftarrow \lambda_i - \alpha\Big(\eta_0 + \sum_j \lambda_j - \nabla A^*(\mathbb{E}_{p_i(z)}[s(z)])\Big), \tag{4}$$

$$\textbf{Outer update:} \quad \theta \leftarrow \eta_0 + \sum_j \lambda_j, \tag{5}$$

---

[1]We use e.g. $p$ to refer to a distribution, and $p(.)$ or $p(z)$ for its density, throughout.

[2]Going forward, we assume $i$ and $j$ to be taken from the index set $\{1, \ldots, m\}$ unless otherwise specified.

[3]This interpretation is not precise, since it is not necessarily the case that $\lambda_i \in \Omega$.

where $p_i \in \mathcal{F}_i$ denotes the $i$-th *tilted distribution*, with density

$$p_i(z) \propto \exp\left((\theta - \beta_i^{-1}\lambda_i)^\top s(z) + \beta_i^{-1}\ell_i(z)\right). \tag{6}$$

The hyperparameter $\alpha$ controls the level of damping, which is used to aid convergence. An undamped inner update (with $\alpha = 1$) can be seen as performing *moment matching* between $p$ and $p_i$. The expectation in (4) is most often computed analytically, or estimated using deterministic numerical methods. It can also be estimated by sampling from $p_i$, however, we will later show that the resulting stochasticity can lead to a biased and unstable procedure. The inner updates can be performed either serially or in parallel (over $i$). In this work we assume they are applied in parallel, but the ideas presented can easily be extended to the serial case. Heskes and Zoeter [25] showed that (4) follows a decrease direction in $L$ with respect to $\lambda_i$, and so is guaranteed to decrease $L$ when $\alpha$ is small enough. See Appendix C for a derivation of the inner update. When $\{\lambda_i\}_i$ are at a partial minimum of $L$ (for fixed $\theta$), the outer update performs an exact partial maximisation of the primal form of the variational problem – see Hasenclever et al. [17] for details. All of the EP variants considered in this paper differ only in their handling of the inner minimisation, and this is our focus.

**Unified EP algorithm**  The double-loop EP algorithm of Heskes and Zoeter [24] repeats the inner update to convergence before performing each outer update, which ensures convergence of the overall procedure. The usual presentation of EP combines (4) and (5) into a single update in $\lambda_i$ (see Appendix B), however, Jylänki et al. [30] observed that it can also be viewed as performing double-loop EP with just a single inner update per outer update (which is not guaranteed to converge in general). By taking this view, we are we are able to present EP, power EP, and their double-loop counterparts as a single algorithm, presented in Algorithm 1. We do so primarily to illustrate how these variants are related to one another, and to the new variants of Section 3.

---

**Algorithm 1** EP ($\beta_i$=1,$n_{\text{inner}}$=1), power EP ($\beta_i{\neq}1$,$n_{\text{inner}}$=1), and their double-loop variants ($n_{\text{inner}}$>1)

**Require:** $\mathcal{F}, \eta_0, \{\beta_i\}_i, \{\ell_i(z)\}_i, \{\lambda_i\}_i, n_{\text{inner}}, \alpha$
  **while** not converged **do**
    $\theta \leftarrow \eta_0 + \sum_j \lambda_j$

> Stochastic estimation of this expectation can lead to biased and unstable updates.

    **for** 1 to $n_{\text{inner}}$ **do**
      **for** $i = 1$ to $m$ *in parallel* **do**
        $\lambda_i \leftarrow \lambda_i - \alpha(\eta_0 + \sum_j \lambda_j - \nabla A^*(\mathbb{E}_{p_i(z)}[s(z)]))$
  **return** $\{\lambda_i\}_i$

---

Note that dependence on $\{\beta_i\}_i$ comes through the definition of $p_i(.)$. We can think of "exact" double-loop EP as corresponding to $n_{\text{inner}} = \infty$, with the inner loop exiting once some convergence criterion has been satisfied. In practice, a truncated inner loop is often used by setting $n_{\text{inner}}$ to some small number [17, 30]. Upon convergence of Algorithm 1, the approximation $p(z) \approx p^*(z)$ is given by (2). Going forwards we will refer to the family of algorithms encompassed by Algorithm 1 simply as EP.

**Stochastic moment estimation**  EP is typically applied to models for which the updates either have closed-form expressions, or can be accurately estimated using deterministic numerical methods. However, as update (4) only depends on the target distribution through expectations under the tilted distributions, this suggests that updates could also be performed using *sampled* estimates of those expectations. By using MC methods to estimate the tilted distribution moments, EP can be used in a black-box manner, dramatically expanding the set of models it can be applied to. Instead of performing a single large sampling task—as would be required by applying MC methods directly—EP can instead solve several simpler ones, gaining significant computational advantages [17, 48, 52]. Unfortunately, when performed naively, update (4) is not robust to MC noise. This is because the moment estimates are converted to the natural (site) parameter space by mapping through $\nabla A^*(.)$, which is not linear in general, and so MC noise in the estimates leads to biased updates of $\lambda_i$.

## 3 Fearlessly stochastic EP algorithms

In this section we show an equivalence between the moment-matching updates of EP and natural gradient descent (NGD). We use this view to motivate two new EP variants which have several advantages when updates are estimated using samples. We conclude with a review of related work.

### 3.1 Natural gradient view of EP

Several EP variants—EP [40, 37], power EP, double loop EP [34], stochastic natural gradient EP [17], and the new ones to follow—differ only in how they perform the *inner* optimisation in (3) with respect to $\{\lambda_i\}_i$. The optimisation can be viewed as one with respect to the $m + 1$ distributions $p, p_1, \ldots, p_m$, which are jointly parameterised by $\{\lambda_i\}_i$ [17]. Natural gradient descent (NGD) [1], which involves preconditioning a gradient with the inverse Fisher information matrix (FIM) of a distribution, is an effective method for optimising parameters of probability distributions. It would therefore seem desirable to apply NGD to the inner optimisation, yet doing so is not straightforward. While the FIM can be naturally extended to the product of statistical manifolds that is the solution space, computing and inverting it will not generally be tractable, making (the natural extension of) NGD in this space infeasible.

Consider instead $\tilde{p}_i^{(t)} \in \mathcal{F}$ with natural parameter $\eta_i^{(t)}(\lambda_i) = \eta_0 + \sum_j \lambda_j^{(t)} + \alpha^{-1}(\lambda_i - \lambda_i^{(t)})$. The $t$ superscript on the site parameters indicates that they are fixed for the current iteration, and so the distribution is fully parameterised by $\lambda_i$. Note that when $\lambda_j = \lambda_j^{(t)} \, \forall \, j$, we have $\tilde{p}_i^{(t)} = p$. Proposition 1, which we prove in Appendix D, states that the moment-matching updates of EP can be viewed as performing NGD in $L$ with respect to *mean parameters* of $\tilde{p}_i^{(t)}$.

We will make use of the following properties: for an exponential family with log partition function $A(.)$, the gradient $\nabla A(.)$ provides the map from natural to mean parameters, and this mapping is one-to-one for minimal families, with the inverse given by $\nabla A^*(.)$. See Appendix A for details.

**Proposition 1.** *For $\alpha > 0$, the moment-matching update of EP (4) is equivalent to performing an NGD step in $L$ with respect to the mean parameters of $\tilde{p}_i^{(t)}$ with step size $\alpha^{-1}$. That is, for $\mu_i = \mathbb{E}_{\tilde{p}_i^{(t)}(z)}[s(z)]$, and $\tilde{F}_i^{(t)}(\mu_i)$ the FIM of $\tilde{p}_i^{(t)}$ with respect to $\mu_i$, we have*

$$\mu_i - \alpha^{-1}\left[\tilde{F}_i^{(t)}(\mu_i)\right]^{-1}\frac{\partial L}{\partial \mu_i} = \left(\nabla A \circ \eta_i^{(t)}\right)\left(\lambda_i - \alpha\left(\eta_0 + \sum_j \lambda_j - \nabla A^*\left(\mathbb{E}_{p_i(z)}[s(z)]\right)\right)\right). \quad (7)$$

Note that the right hand side of (7) just maps update (4) to mean parameters of $\tilde{p}_i^{(t)}$. [4] We discussed in Section 2 that noise in the moment estimates results in bias in the update to $\lambda_i$ of EP, due to the noise being passed through the nonlinear map $\nabla A^*(.)$. This alone would not be a problem if the updates followed unbiased descent direction estimates in some other fixed parameterisation. Let $\eta_i = \nabla A^*(\mu_i)$, then, from the definition of $L$ and $\mu_i$, we have

$$\frac{\partial L}{\partial \mu_i} = \frac{\partial \eta_i}{\partial \mu_i}\frac{\partial \lambda_i}{\partial \eta_i}\left(\nabla A\left(\eta_0 + \sum_j \lambda_j\right) - \mathbb{E}_{p_i(z)}[s(z)]\right), \quad (8)$$

and so when $\mathbb{E}_{p_i(z)}[s(z)]$ is estimated with unbiased samples, the NGD update (7) in $\mu_i$ is also unbiased. However, a *sequence* of updates introduces bias. This is because the parameterisation changes from one update to the next, with the mean parameters of $\tilde{p}_i^{(t)}$ mapped to those of $\tilde{p}_i^{(t+1)}$ by

$$\mu_i^{(t+1)} = \left(\nabla A \circ \eta_i^{(t+1)} \circ \left(\eta_i^{(t)}\right)^{-1} \circ \nabla A^*\right)\left(\mu_i^{(t)}\right), \quad (9)$$

where $(\eta_i^{(t)})^{-1}(.)$ is the inverse of $\eta_i^{(t)}(.)$, and $\eta_i^{(t+1)}(.)$ is defined using the site parameters *after* the update at time $t$. This map is not linear in general, and so an unbiased, noisy update of $\mu_i$ at one time step, results in bias when mapped to the next. This bias can make the EP updates unstable in the presence of MC noise, leading previous work to use methods, specific to $\mathcal{F}$, for obtaining approximately unbiased *natural* parameter estimates from samples. However, even with these debiasing methods, a relatively large number of samples is still needed in practice [48, 52]. Here we have assumed parallel application of (4), but the bias at site $i$ is induced whenever the parameters for other sites change from one update to the next, which occurs in both parallel and serial settings.

We now use the natural gradient interpretation of the EP updates to motivate two new variants that are far more robust to MC noise. In particular, they are stable, and most sample-efficient, when updates are estimated with just a single sample. Furthermore, unlike methods that rely on deibasing estimators, they do not require sample thinning (see Section 3.4). This largely eliminates two hyperparameters for the practitioner. While practical considerations, such as ensuring efficient use of parallel hardware, may justify using more than one sample per update, this can in principle be tuned *a priori*, much like the batch size hyperparameter of stochastic gradient descent.

---

[4]Note the apparent contradiction that by decreasing $\alpha$ (increasing the damping) we actually *increase* the NGD step size. This is resolved by observing that the definition of $\tilde{p}_i^{(t)}$ also changes with $\alpha$.

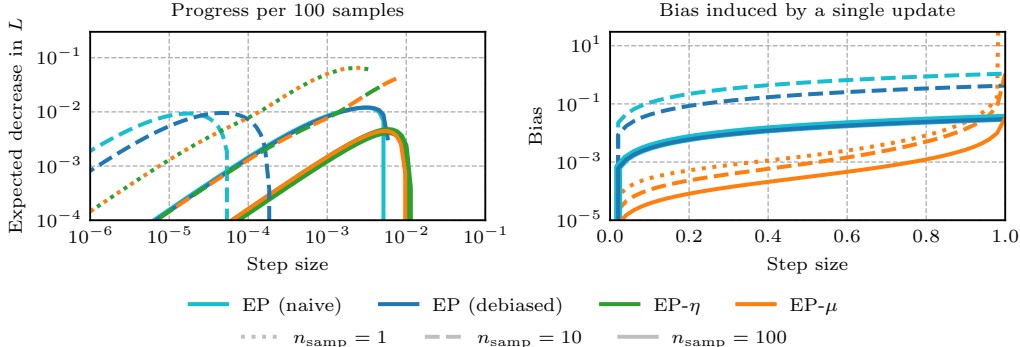

Figure 1: The effect of step size ($\alpha$ or $\epsilon$) and number of MC samples ($n_{\text{samp}}$) on different EP variants in a stochastic version of the clutter problem of Minka [37]. EP (naive) uses maximum likelihood estimation for the updates, and EP (debiased) uses the estimator of Xu et al. [52]. Step size corresponds to $\alpha$ for EP, and $\epsilon$ for EP-$\mu$ and EP-$\eta$. Only EP-$\eta$ and EP-$\mu$ can perform 1-sample updates, hence the other traces are not visible. The left panel shows the expected decrease in $L$ after $100/n_{\text{samp}}$ steps. Performing e.g. $100\times$ 1-sample steps, or $10\times$ 10-sample steps, achieves a much larger decrease in $L$ than a single 100-sample step. The right panel shows the magnitude of the bias in $\lambda_i$ after a single parallel update, averaged over all sites and dimensions. The bias of EP-$\mu$ shrinks far faster as the step size decreases than that of EP. EP-$\eta$ is always unbiased and so is not visible.

## 3.2  EP-$\eta$

We showed in the previous section that bias is introduced into the sequence of NGD updates due to a nonlinear map from one parameterisation to the next. If the map were affine, the sequence would remain unbiased. This can be achieved by performing NGD with respect to $\eta_i$, the *natural* parameters of $\tilde{p}_i^{(t)}$. Then, the map from one parameterisation to the next (by equating $\lambda_i$) is given by $\eta_i^{(t+1)} \circ (\eta_i^{(t)})^{-1}$, which is linear. The NGD direction in $L$ with respect to $\tilde{p}_i^{(t)}$ and $\eta_i$ is given by

$$-\left[\tilde{F}_i(\eta_i)\right]^{-1}\frac{\partial L}{\partial \eta_i} = -\alpha \frac{\partial \eta_i}{\partial \mu_i}\Big(\nabla A\big(\eta_0 + \sum\nolimits_j \lambda_j\big) - \mathbb{E}_{p_i(z)}[s(z)]\Big). \tag{10}$$

See Appendix E for a derivation. Note that $\alpha$ plays a similar role as the NGD step size in this parameterisation, but it also affects $\tilde{p}_i^{(t)}$, and so to obtain a more faithful NGD interpretation we fix $\alpha = 1$ and introduce an explicit NGD step size $\epsilon$ to decouple from the effect on $\tilde{p}_i^{(t)}$. The resulting update can be expressed directly as an update in $\lambda_i$, by applying $(\eta_i^{(t)})^{-1}$ to the updated $\eta_i$, giving

$$\lambda_i \leftarrow \lambda_i - \epsilon \frac{\partial \eta_i}{\partial \mu_i}\Big(\nabla A\big(\eta_0 + \sum\nolimits_j \lambda_j\big) - \mathbb{E}_{p_i(z)}[s(z)]\Big). \tag{11}$$

We can use automatic differentiation to efficiently compute (11), by recognising the second term as a Jacobian-vector product (JVP) with respect to $\eta_i(\mu_i) = \nabla A^*(\mu_i)$.[5] In summary, by performing NGD with respect to *natural* parameters of $\tilde{p}_i^{(t)}$, instead of mean parameters, the resulting sequence of updates is unbiased in $\lambda_i$. We call the resulting procedure—which is given in Algorithm 2—EP-$\eta$, to emphasise that we have simply changed the NGD parameterisation of EP. The unbiased updates of EP-$\eta$ allow it to be more sample-efficient than EP by using a smaller number of samples per iteration. The left panel of Figure 1 demonstrates this effect in a stochastic version of the clutter problem [37].

## 3.3  EP-$\mu$

The bias introduced by the updates of EP cannot be mitigated by reducing $\alpha$ (increasing the damping) without also proportionally sacrificing the amount of progress made by each update. More concretely, let the bias in dimension $d$ of $\mu_i^{(t+1)}$, after an update at time $t$, be defined as $\mathbb{E}[\mu_i^{(t+1)} - \bar{\mu}_i^{(t+1)}]_d$, where $\bar{\mu}_i^{(t+1)}$ is the value of $\mu_i^{(t+1)}$ after a *noise-free* update, and expectation is taken over the sampling distributions of all parallel updates at time $t$. Then, Proposition 2 below, which we prove in Appendix G, summarises the effect of decreasing $\alpha$ on EP.

---

[5]This can be computed using either forward *or* reverse mode automatic differentiation, as $\frac{\partial \eta_i}{\partial \mu_i}$ is symmetric.

**Proposition 2.** *After update* (4) *is executed in parallel over $i$, as $\alpha \to 0^+$, both the expected decrease in $L$, and the bias $\mathbb{E}[\mu_i^{(t+1)} - \bar{\mu}_i^{(t+1)}]_d$, are $O(\alpha)$ for all $d$.*

Proposition 1 at the beginning of this section states that update (4) can be viewed as performing an NGD step with respect to $\mu_i$ with a step size of $\alpha^{-1}$. However, changing $\alpha$ also has the effect of changing the definition of $\tilde{p}_i^{(t)}$. It is then natural to wonder what happens if we fix $\alpha = 1$ and introduce an explicit step size for NGD, $\epsilon$, resulting in the update

$$\mu_i \leftarrow \mu_i - \epsilon\Big(\nabla A\big(\eta_0 + \sum_j \lambda_j\big) - \mathbb{E}_{p_i(z)}[s(z)]\Big). \tag{12}$$

See Appendix F for a derivation. It turns out that in doing so, when we decrease $\epsilon$, the bias shrinks far faster than the expected decrease in $L$.

**Proposition 3.** *After update* (12) *is executed in parallel over $i$, as $\epsilon \to 0^+$, the expected decrease in $L$ is $O(\epsilon)$, and the bias $\mathbb{E}\left[\mu_i^{(t+1)} - \bar{\mu}_i^{(t+1)}\right]_d$ is $O(\epsilon^2)$, for all $d$.*

Proposition 3, which we prove in Appendix G, tells us that by using update (12), we can reduce the bias to arbitrarily small levels while still making progress in decreasing $L$. Update (12) can also be expressed directly as an update in $\lambda_i$ by applying $(\eta^{(t)})^{-1} \circ \nabla A^*$, giving

$$\lambda_i \leftarrow \nabla A^*\Big((1-\epsilon)\nabla A\big(\eta_0 + \sum_j \lambda_j\big) + \epsilon\mathbb{E}_{p_i(z)}[s(z)]\Big) - \eta_0 - \sum_{j \neq i}\lambda_j, \tag{13}$$

which has the simple interpretation of performing EP, but with damping of the moments instead of the site (natural) parameters. In summary, if we perform NGD with respect to the same mean parameterisation as EP, but treat the NGD step size as a free parameter, $\epsilon$, we can obtain updates that are *approximately* unbiased while still making progress. We call this variant EP-$\mu$, again to indicate that we are simply performing the NGD update of EP in the mean parameter space.[6] The resulting procedure is also given by Algorithm 2.

---

**Algorithm 2** EP-$\eta$ and EP-$\mu$ (differences with Algorithm 1 are highlighted in green)

---
**Require:** $\mathcal{F}, \eta_0, \{\beta_i\}_i, \{\ell_i(z)\}_i, \{\lambda_i\}_i, n_{\text{inner}}, \epsilon$
   **while** not converged **do**
      $\theta \leftarrow \eta_0 + \sum_j \lambda_j$
      **for** 1 to $n_{\text{inner}}$ **do**
         **for** $i = 1$ to $m$ *in parallel* **do**
            update $\lambda_i$ using (11) for EP-$\eta$, or (13) for EP-$\mu$
   **return** $\{\lambda_i\}_i$

---

The computational cost of EP-$\mu$ is lower than that of EP-$\eta$ because it does not require JVPs through $\nabla A^*(.)$.[7] The drawback is that the updates of EP-$\mu$ do still retain some bias, however, we find that the bias of EP-$\mu$ typically has negligible impact on its performance relative to EP-$\eta$. This is evident in the clutter problem of Minka [37], as demonstrated in the left panel of Figure 1, as well as in the larger scale experiments of Section 4. The right panel of Figure 1 illustrates how quickly the bias in $\lambda_i$ shrinks as the step size of EP-$\mu$ is decreased. Note that the results of this subsection are stated in terms of bias in $\mu_i$, but it is straightforward to show that equivalent results also hold for $\lambda_i$ using Taylor series arguments.

### 3.4 Related work

The link between natural gradients and moment matching in an exponential family is well known [6], but the connection with EP shown here – which relies on identification of the distribution $\tilde{p}_i^{(t)}$ and parameterisation $\eta_i^{(t)}$ – is new, to the best of our knowledge. Bui et al. [11] showed that the updates of (power) EP are equivalent to performing NGD in a local variational free energy objective when taking the limit of the power parameter as $\beta_i \to \infty$, and Wilkinson et al. [51] showed that this extends to the (global) variational free energy objective for models with a certain structure. Our result is different in that it relates to NGD of the EP variational objective, and applies for all values of $\beta_i$.

---

[6]EP too is performing NGD in mean parameters, but using a "step size" that also affects the distribution $\tilde{p}_i$.
[7]See Appendix I for a detailed discussion of the computational costs of EP-$\eta$ and EP-$\mu$.

EP was combined with Markov chain Monte Carlo (MCMC) moment estimates by Xu et al. [52] in a method called *sampling via moment sharing* (SMS), and later by Vehtari et al. [48] in the context of hierarchical models. In both works, when $\mathcal{F}$ was multivariate normal (MVN) —arguably the most useful case—the authors used an estimator for the updates that is unbiased when $\tilde{p}_i$ is also MVN. Although this estimator is only *approximately* unbiased in general, it can help to stabilise the updates significantly. Even so, it is necessary to use a relatively large number of samples for the updates, with several hundred or more being typical. Using many samples per update is inefficient, as the update direction can change from one iteration to the next. Furthermore, such estimators typically rely on a known number of *independent* samples. Samples drawn using MCMC methods are generally autocorrelated, necessitating the use of sample thinning before the estimators can be applied. This adds another hyperparameter to the procedure—the thinning ratio—and it is also inefficient due to the discarding of samples. The practitioner is required to choose the number of samples, the thinning ratio, and the amount of damping, all of which affect the accuracy, stability, and computational efficiency of the procedure. There is no easy way to choose these *a priori*, forcing the practitioner to either set them conservatively (favouring accuracy and stability), or to find appropriate settings by trial and error, both of which are likely to expend time and computation unnecessarily.

The stochastic natural gradient EP (SNEP) method of Hasenclever et al. [17], which is closely related to our work, also optimises the EP variational objective using a form of NGD. The SNEP updates are unbiased in the presence of MC noise, allowing them to be performed with as little as one sample, without relying on $\mathcal{F}$-specific debiasing estimators, or the sample thinning that would entail. In SNEP, NGD is performed with respect to mean parameters of the *site potentials*, which are treated as bona fide distributions in $\mathcal{F}$. In contrast, we showed that EP can already be viewed as performing NGD, but with respect to the distributions $\{\tilde{p}_i^{(t)}\}_i$, and our new variants, EP-$\eta$ and EP-$\mu$, are able to gain the same advantages as SNEP, but using the same distributions for NGD as EP. Hasenclever et al. [17] showed that in some settings, SNEP can obtain accurate point estimates fairly rapidly. However, we find that it typically converges far slower than both EP and our new variants, consistent with findings in Vehtari et al. [48]. We argue that this is because the site potentials (when considered as distributions) bear little resemblance with the distributions that are ultimately being optimised, and so their geometry is largely irrelevant for the optimisation of $L$. In contrast, the geometry of $p_i^{(t)}$ is closely related to that of $L$, as we show in Appendix H.

We note that Xu et al. [52] and Hasenclever et al. [17] also proposed methods for performing updates in an asynchronous fashion, significantly reducing the frequency and cost of communication between nodes in distributed settings. In principle, these methods could be combined with those presented in this paper, but we do not consider them further here.

## 4 Evaluation

In this section we demonstrate the efficacy of EP-$\eta$ and EP-$\mu$ on a variety of probabilistic inference tasks. In each experiment, the task was to perform approximate Bayesian inference of unobserved parameters $z$, given some observed data $\mathcal{D}$. All of the models in these experiments followed the same general structure, consisting of a minimal exponential family prior over $z$, $p_0$, and a partition of the data $\{\mathcal{D}_i\}_i^m$, where each block $\mathcal{D}_i$ depends on both $z$ and an additional vector of *local* latent variables $w_i$ that also depend on $z$. That is, the joint density has the form

$$p_0(z)\prod_i p(w_i \mid z)p(\mathcal{D}_i \mid w_i, z),\tag{14}$$

where $p_0 \in \mathcal{F}$. This structure is shown graphically in Appendix J. To perform approximate inference of $z$, we first define $\ell_i(z)$ as the log likelihood of $\mathcal{D}_i$ given $z$, with $w_i$ marginalised out. That is,

$$\ell_i(z) = \log \int p(\mathcal{D}_i, w_i \mid z)\mathrm{d}w_i.\tag{15}$$

Then, given $p_0(z)$ and $\{\ell_i(z)\}_i$, we define $p^*(z)$ as (1), and proceed to find an approximation $p(z) \approx p^*(z)$ using the methods described in earlier sections. Note that sampling from the tilted distribution $p_i(z)$ requires jointly sampling over $z$ and $w_i$ and taking the marginal. EP is a particularly appealing framework for performing inference in this setting, as the dimensionality of the sampled distributions is constant with respect to $m$, mitigating the curse of dimensionality experienced by conventional MCMC approaches [48]. In our experiments we used NUTS [27] to perform the sampling, consistent with prior work [48, 52]. In each experiment we compared EP-$\eta$ and EP-$\mu$ with

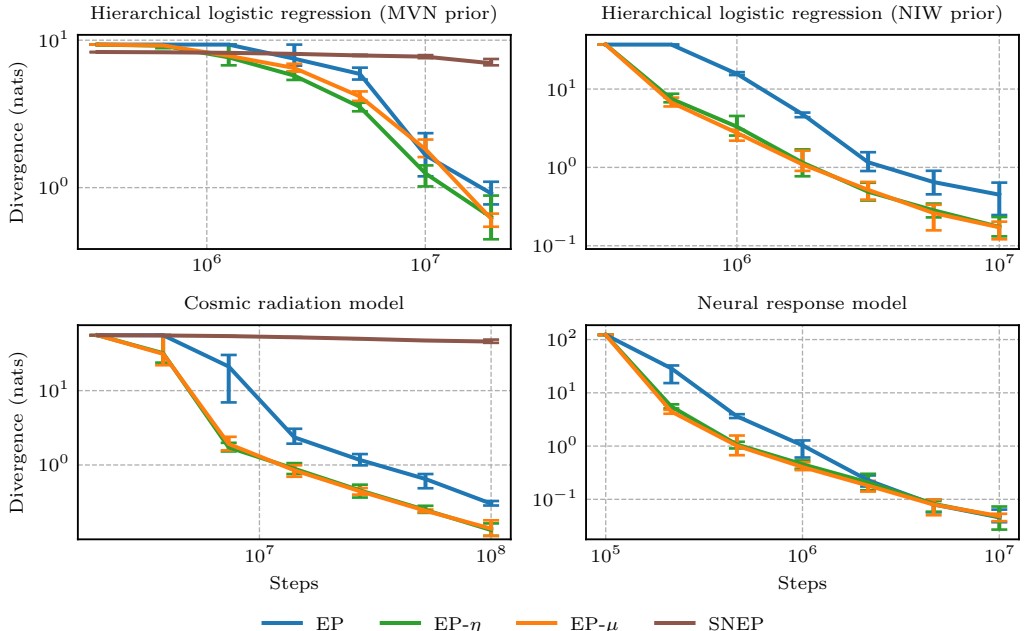

Figure 2: Pareto frontiers showing the number of NUTS steps ($x$-axis) against the KL divergence from $p$ to an estimate of the optimum ($y$-axis). Each point on the plot marks the lowest average KL divergence attained by *any* hyperparameter setting by that step count. Error bars mark the full range of values for the marked hyperparameter setting across 5 random seeds.

EP, in the manner used by Xu et al. [52] and Vehtari et al. [48]. When $\mathcal{F}$ was MVN we also compared to SNEP [17]. For the models with normal-inverse Wishart (NIW) $\mathcal{F}$, we were unable to find an initialisation for SNEP that would allow us to perform a meaningful comparison. We discuss this point further in Appendix J.

To evaluate the performance of the different variants, we monitored KL divergence to an *estimate* of the optimum, obtained by running EP with a large number of samples. We used 500 different hyperparameter settings for each variant, chosen using random search, and repeated each run 5 times using different random seeds for NUTS. All runs of EP-$\eta$ and EP-$\mu$ were performed with $n_{\mathrm{samp}} = 1$ and $n_{\mathrm{inner}} = 1$. The results of our evaluation are summarised by the Pareto frontiers in Figure 2.

We see that EP-$\eta$ and EP-$\mu$ can typically reach a given level of accuracy (distance from an estimate of the optimum) faster than EP, often significantly so. We stress that the frontiers for EP-$\eta$ and EP-$\mu$ are traced out with $n_{\mathrm{samp}}$ and $n_{\mathrm{inner}}$ each fixed to 1, with no sample thinning, varying only the step size hyperparameter $\epsilon$, with higher accuracy/cost regions of the frontier corresponding to lower values of $\epsilon$. In contrast, to trace out the frontier of EP we must jointly vary $\alpha$, $n_{\mathrm{samp}}$, and the thinning ratio. See Appendix K for examples of this effect. The performances of EP-$\eta$ and EP-$\mu$ are very similar. We found SNEP to be significantly slower than the other methods, consistent with findings by Vehtari et al. [48]. Pareto frontiers with respect to wall-clock time can be found in Appendix L. Code for these experiments can be found at `https://github.com/cambridge-mlg/fearless-ep`. We now provide a brief overview of individual experiments, with further details given in Appendix J.

**Hierarchical logistic regression with MVN prior** Hierarchical logistic regression (HLR) is used to perform binary classification in a number of groups when the extent to which data should be pooled across groups is unknown [15]. We performed approximate Bayesian inference in a HLR model using a MVN prior over the group-level parameters. We applied this model to synthetic data, generated using the same procedure as Vehtari et al. [48], which was designed to be challenging for EP. In this experiment $z \in \mathbb{R}^8$, $w_i \in \mathbb{R}^4 \ \forall i$, and each of the $m = 16$ groups had $n = 20$ observations. In the upper-left panel of Figure 2 we see that EP-$\eta$ and EP-$\mu$ typically reach a given level of accuracy faster than EP and SNEP.

**Hierarchical logistic regression with NIW prior**    We also performed approximate Bayesian inference in a similar HLR model, but using a normal-inverse Wishart (NIW) prior over group-level parameters, allowing for correlation of regression coefficients within groups. We applied this model to political survey data, where each of $m = 50$ groups corresponded to responses from a particular US state, with 7 predictor variables corresponding to characteristics of a given survey respondent, so that $w_i \in \mathbb{R}^7 \ \forall \ i$. There were $n = 97$ survey respondents per state. The data, taken from the 2018 Cooperative Congressional Election Study, related to support for allowing employers to decline coverage of abortions in insurance plans [33, 43]. The upper-right panel of Figure 2 shows that EP-$\eta$ and EP-$\mu$ consistently reach a given level of accuracy significantly faster than EP.

**Cosmic radiation model**    A hierarchical Bayesian model was used by Vehtari et al. [48] to capture the nonlinear relationship between diffuse galactic far ultraviolet radiation and 100-μm infrared emission in various sectors of the observable universe, using data obtained from the Galaxy Evolution Explorer telescope. In this model each $w_i \in \mathbb{R}^9$ parameterised a nonlinear regression model using data obtained from one of $m$ sections of the observable universe. The $m$ regression problems were related through hyperparameters $z \in \mathbb{R}^{18}$, which parameterised the section-level parameter densities. The prior, $p_0 \in \mathcal{F}$, was MVN. The specifics of the nonlinear regression model are quite involved and we refer the reader to Vehtari et al. [48] for details. We were unable to obtain the dataset, and so we generated synthetic data using parameters that were tuned by hand to try and match the qualitative properties of the original data – see Appendix N for examples. We used a reduced number of $m = 36$ sites and $n = 200$ observations per site to allow us to perform a comprehensive hyperparameter search. The lower-left panel of Figure 2 shows once again that EP-$\eta$ and EP-$\mu$ consistently reach a given level of accuracy significantly faster than EP and SNEP.

**Neural response model**    A common task in neuroscience is to model the firing rates of neurons under various conditions. We performed inference in a hierarchical Bayesian neural response model, using recordings of V1 complex cells in an anesthesised adult cat [4]. In this dataset, 10 neurons in a specific area of cat V1 were simultaneously recorded under the presentation of 18 different visual stimuli, each repeated 8 times, for a total of 144 trials. We modelled the observed spike counts of the 10 neurons in each trial as Poisson, with latent (log) intensities being MVN, with mean and covariance (collectively $z \in \mathbb{R}^{65}$) drawn from a NIW prior $p_0 \in \mathcal{F}$. Inference of $z$ amounted to inferring the means, and inter-neuron covariances, of latent log firing rates across stimuli. We grouped the data into $m = 8$ batches of $n = 18$ trials each, so that the latent log firing rates for all trials in batch $i$ were jointly captured by $w_i \in \mathbb{R}^{180}$. The lower-right panel of Figure 2 shows that EP-$\eta$ and EP-$\mu$ either match or beat the speed of EP for reaching any given level of accuracy. For high levels of accuracy the performances of the methods appear to converge with one another, but note that the asymptote is not exactly zero because the optimal solution was estimated by running EP with a large (but finite) number of samples.

## 5    Limitations

In this work we have largely assumed that the drawing of MC samples is the dominant cost. In practice, other costs become relevant, which may shift the balance in favour of using more samples to estimate updates. We discuss this in Appendix I, along with strategies for reducing computational overheads. We note, however, that wall-clock time results for our evaluation experiments (in Appendix L) are in line with the results of Section 4, indicating that the drawing of MC samples was indeed the dominant cost.

When EP and its variants—including those introduced here—are combined with MC methods, the underlying samplers typically have hyperparameters of their own, which are often adapted using so-called warmup phases. While EP-$\eta$ and EP-$\mu$ significantly reduce the complexity of tuning hyperparameters specific to EP, they do not help with those of the underlying samplers. This means that the complexity of hyperparameter tuning (for all EP variants) is still greater than that for direct MC methods. We used comprehensive hyperparameter searches in our experiments in order to perform a meaningful comparison with baseline methods, necessarily limiting the scale of problems we could tackle.

Our focus has been on developing improved methods for performing EP when the updates must be estimated with noise, with the motivation for this kind of setting having been discussed by prior

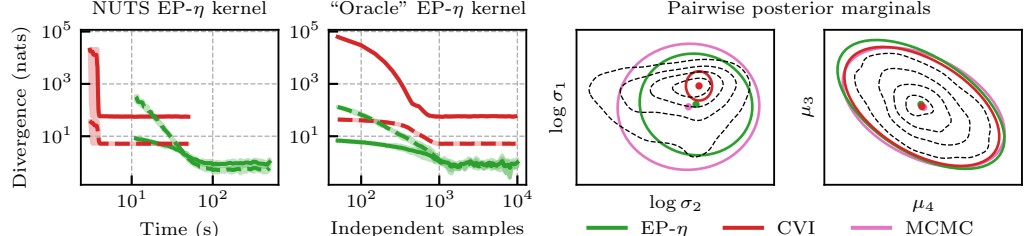

Figure 3: Comparison of EP-$\eta$ with conjugate-computation variational inference (CVI) on a hierarchical logistic regression model. The two leftmost plots show forward (solid) and reverse (dashed) KL divergences between the approximation of each method and a MVN distribution estimated directly from MCMC samples. The left panel shows a comparison with respect to wall-clock time, when NUTS is used as the underlying sampling kernel for EP-$\eta$. The left-middle panel shows a similar comparison, but with respect to the number of samples drawn, and using an "oracle" sampling kernel for EP-$\eta$. Hyperparameters were tuned for each method and shaded regions show the range of trajectories across 5 random seeds. The right and right-middle panels show pairwise marginals of the various MVN approximations overlaid on contours of the true posterior. Coloured dots and ellipses correspond to means and 2-standard-deviation contours, respectively. See Section 5 for discussion of these results, and Appendix M for further details.

work [52, 17, 48]. EP and its variants are agnostic to the choice of sampling kernel, and as such we have not discussed issues relating to the performance of the underlying samplers. In the left panel of Figure 3 however, we highlight the relative inefficiency of using EP-$\eta$ with NUTS as the underlying sampler when compared to conjugate-computation variational inference (CVI), an efficient variational inference method [31]. While EP-$\eta$ is able to obtain much more faithful posterior approximations than CVI, it is significantly slower when measured in wall-clock time, despite the gains made over its predecessors. However, this is due to the relative cost of drawing independent samples with NUTS, as the sample efficiency of EP-$\eta$ is roughly equivalent to that of CVI when an (illustrative) "oracle" sampling kernel is used, as demonstrated in the left-middle panel of Figure 3. Further discussion and details of the experiments behind Figure 3 can be found in Appendix M.

## 6 Future work

In Section 5, we demonstrated that the efficiency of EP variants is constrained by that of of the underlying samplers. There are several ways in which their performance could be improved by using knowledge of $p$ to guide sampling from $p_i$. One such improvement would be to use $p$ to set MCMC hyperparameters directly, obviating the need to adapt them during warm-up phases, e.g. when $p \in \mathcal{F}$ is MVN, its precision matrix could be used directly as the mass matrix in Hamiltonian Monte Carlo schemes. Another potential improvement is that samples from $p$ could be used to re-initialise MCMC chains at each iteration, providing approximate independence between updates for a small additional cost. It would also be worthwhile considering how our methods can be efficiently implemented on modern compute hardware to better take advantage of the inherently parallel nature of EP, e.g. practical performance could be improved by using sampling kernels that are themselves designed for efficient parallel execution [26]. Performance may be further improved in distributed settings by using asynchronous extensions of EP in order to minimise communication-related overheads and delays [17, 52]. We leave investigation of these ideas to future work.

## 7 Conclusion

In this work, we used a novel interpretation of the moment-matching updates of EP to motivate two new EP variants that are far more robust to MC noise. We demonstrated that these variants can offer an improved speed-accuracy trade-off compared to their predecessors, and are easier to tune.

# 8 Acknowledgements

We would like to thank the anonymous reviewers for providing valuable feedback, and Jihao Andreas Lin for sharing his plotting expertise. Jonathan So is supported by the University of Cambridge Harding Distinguished Postgraduate Scholars Programme. Richard E. Turner is supported by Google, Amazon, ARM, Improbable, EPSRC grant EP/T005386/1, and the EPSRC Probabilistic AI Hub (ProbAI, EP/Y028783/1).

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

# A Exponential families of distributions

In this appendix, we give a very brief overview of exponential families of distributions. This is largely a condensed summary of relevant material from Wainwright and Jordan [49], and we refer the reader to the original work for a far more detailed treatment.

The exponential family of distributions $\mathcal{F}$, defined by the $d$-dimensional, vector-valued sufficient statistic function $s(.)$, and base measure $\nu(.)$, has density function

$$p(z) = \exp(\eta^\top s(z) - A(\eta)), \tag{16}$$

taken with respect to $\nu(.)$, where $A(\eta) = \log \int \exp(\eta^\top s(z)) \mathrm{d}\nu(z)$ is known as the *log-partition* function. $\eta \in \Omega$ are known as the *natural parameters*, and are used to index a specific member of $\mathcal{F}$. $\Omega$ is the set of normalisable natural parameters of $\mathcal{F}$, given by

$$\Omega = \left\{ \eta \in \mathbb{R}^d \;\middle|\; \int \exp(\eta^\top s(z)) \mathrm{d}\nu(z) < \infty \right\}, \tag{17}$$

and is known as the *natural domain* of $\mathcal{F}$. $\Omega$ is a convex set, and when it is open, the family $\mathcal{F}$ is said to be *regular* – we shall only consider regular families in this work. When the components of $s(.)$ are linearly independent, $\mathcal{F}$ is said to be *minimal*. For minimal families, each distribution in the family is associated with a unique natural parameter vector $\eta$. An exponential family that is not minimal is said to be *overcomplete*, in which case, each distribution is associated with an entire affine subspace of $\Omega$.

The expected sufficient statistics of a distribution with density $p(.)$ with respect to base measure $\nu(.)$, are given by $\mathbb{E}_{p(z)}[s(z)]$. Let $\mathcal{M}$ be defined as the set of expected sufficient statistics that can be attained by *any* density $p(.)$ with respect to base measure $\nu(.)$ – that is,

$$\mathcal{M} = \left\{ \mu \in \mathbb{R}^d \;\middle|\; \exists p(.) : \int p(z) s(z) \mathrm{d}\nu(z) = \mu \right\}. \tag{18}$$

All vectors in the interior of $\mathcal{M}$, $\mathcal{M}^\circ$, are realisable by a member of $\mathcal{F}$, and so $\mu \in \mathcal{M}^\circ$ provides an alternative parameterisation of $\mathcal{F}$, in which $\mu$ are known as the *mean parameters*, and $\mathcal{M}^\circ$ is known as the *mean domain*,.

For minimal families, $A(.)$ is strictly convex on $\Omega$. The convex dual of $A(.)$ is defined as

$$A^*(\mu) = \sup_{\eta \in \Omega} \eta^\top \mu - A(\eta), \tag{19}$$

and on $\mathcal{M}^\circ$, $A^*(\mu)$ is equal to the negative entropy of the member of $\mathcal{F}$ with mean parameter $\mu$.

The mean parameters of the member of $\mathcal{F}$ with natural parameter $\eta$ can be obtained by $\mu = \nabla A(\eta)$. For minimal families, this mapping is one-to-one, and the reverse map is given by $\eta = \nabla A^*(\mu)$. For this reason, $\nabla A(.)$ and $\nabla A^*(.)$ are sometimes referred to as the *forward mapping* and *backward mapping* respectively. In this work, we say that a family $\mathcal{F}$ is *tractable* if both the forward and backward mappings can be evaluated efficiently.

The Fisher information matrix (FIM) of an exponential family distribution, with respect to natural parameters $\eta$, is given by, $F(\eta) = \nabla^2 A(\eta)$. Furthermore, in minimal families, the FIM with respect to mean parameters $\mu$ is given by $F(\mu) = \nabla^2 A^*(\mu)$. Using the forward and backward mappings, we also have

$$F(\eta) = \frac{\partial \mu}{\partial \eta} \tag{20}$$

$$F(\mu) = \frac{\partial \eta}{\partial \mu}, \tag{21}$$

and from the inverse function theorem,

$$F(\eta)^{-1} = \frac{\partial \eta}{\partial \mu} \tag{22}$$

$$F(\mu)^{-1} = \frac{\partial \mu}{\partial \eta}. \tag{23}$$

## B  Conventional presentation of EP updates

In this appendix we give a more conventional presentation of the EP updates, and show equivalence with the atypical presentation of Section 2. We will assume here that $\beta_i = 1 \; \forall \; i$, but similar reasoning can be used to show equivalence in the power EP case, where $\beta_i \neq 1$. Note that the update of power EP is often presented as having scale proportional to $\beta_i$, with a separate damping parameter. We subsume both into $\alpha$, with $\alpha = 1$ corresponding to the default damping suggested by Minka [34].

Given a target distribution with the form

$$p^*(z) \propto \exp\left(\eta_0^\top s(z)\right) \prod_{i=1}^{m} \exp\left(\ell_i(z)\right), \tag{24}$$

EP attempts to find an approximation $p \in \mathcal{F}$ such that

$$p(z) \propto \exp\left(\eta_0^\top s(z)\right) \prod_i \exp\left(\lambda_i^\top s(z)\right) \approx p^*(z). \tag{25}$$

The $i$-th *site parameter* $\lambda_i$ can loosely be interpreted as the natural parameters of a $\mathcal{F}$-approximation to the $i$-th target factor, $\exp(\ell_i(z))$. EP is also often used to approximate the normalising constant of $p^*(z)$ – we have omitted details here, but see e.g. Bishop [3].

In order to update $\lambda_i$, EP performs a KL-divergence minimisation between $p(z)$ and a *local* approximation to the target distribution, consisting of just one "real" factor and $(m - 1)$ approximate ones. Specifically,

$$\lambda_i \leftarrow \arg\min_{\lambda_i} \mathrm{KL}[\bar{p}_i(z) \,\|\, p(z; \lambda_i)], \tag{26}$$

where

$$\bar{p}_i(z) \propto \exp\left(\ell_i(z)\right) \prod_{j \neq i} \exp\left(\lambda_j^\top s(z)\right), \tag{27}$$

and we have used the notation $p(z; \lambda_i)$ to make the dependence on $\lambda_i$ explicit. It can be easily shown that the solution to (26) is found by moment matching. That is, the KL-divergence is minimised when

$$\mathbb{E}_{p(z;\lambda_i)}[s(z)] = \mathbb{E}_{\bar{p}_i(z)}[s(z)]. \tag{28}$$

For a minimal exponential family $\mathcal{F}$, there is a unique member of $\mathcal{F}$ with mean parameters $\mathbb{E}_{\bar{p}_i(z)}[s(z)]$, and its natural parameters are given by $\nabla A^*(\mathbb{E}_{\bar{p}_i(z)}[s(z)])$. By equating this with the parameters of $p(z)$, we have

$$\lambda_i = \nabla A^*(\mathbb{E}_{\bar{p}_i(z)}[s(z)]) - \eta_0 - \sum_{j \neq i} \lambda_j. \tag{29}$$

Update (29) is the "standard" update of EP, and can be performed either serially or in parallel (over $i$). Often a damping factor $\alpha$ is used, giving

$$\lambda_i = (1 - \alpha)\lambda_i + \alpha\left(\nabla A^*(\mathbb{E}_{\bar{p}_i(z)}[s(z)]) - \eta_0 - \sum_{j \neq i} \lambda_j\right). \tag{30}$$

In Section 2 we presented EP and several variants as performing some number of "inner" updates, to decrease $L$ with respect to $\lambda_i$, with an "outer" update with respect to $\theta$. These updates are given by

$$\textbf{Inner update:} \quad \lambda_i \leftarrow \lambda_i - \alpha\left(\eta_0 + \sum_j \lambda_j - \nabla A^*(\mathbb{E}_{p_i(z)}[s(z)])\right), \tag{31}$$

$$\textbf{Outer update:} \quad \theta \leftarrow \eta_0 + \sum_j \lambda_j. \tag{32}$$

Immediately after an outer update we have that $\theta = \eta_0 + \sum_j \lambda_j$, and so for $\beta_i = 1$ we have

$$\begin{aligned}
p_i(z) &\propto \exp\left((\theta - \lambda_i)^\top s(z) + \ell_i(z)\right) \\
&= \exp\left(\left(\eta_0 + \sum_{j \neq i} \lambda_j\right)^\top s(z) + \ell_i(z)\right) \\
&\propto \bar{p}_i(z).
\end{aligned} \tag{33}$$

We stated in Section 2 that (non-double-loop) EP can be viewed as performing a single inner update per outer update, and so by rolling (31) and (32) into a single update for $\lambda_i$, we have

$$\begin{aligned}
\lambda_i &\leftarrow \lambda_i - \alpha\left(\eta_0 + \sum_j \lambda_j - \nabla A^*(\mathbb{E}_{\bar{p}_i(z)}[s(z)])\right) \\
&= (1 - \alpha)\lambda_i + \alpha\left(\nabla A^*(\mathbb{E}_{\bar{p}_i(z)}[s(z)]) - \eta_0 - \sum_{j \neq i} \lambda_j\right),
\end{aligned} \tag{34}$$

which is identical to (30).

## C EP inner update derivation

Begin by taking gradients of (3), with respect to $\lambda_i$, giving

$$\nabla_{\lambda_i} L(\theta, \lambda_1, \ldots, \lambda_m) = \nabla A\left(\eta_0 + \sum_j \lambda_j\right) + \beta_i \nabla_{\lambda_i} A_i((\theta - \beta_i^{-1}\lambda_i, \beta_i^{-1})). \qquad (35)$$

At a minimum of $L$ with respect to $\lambda_i$, both sides of (35) must be equal to zero. We can use this to define a fixed-point condition with respect to $\lambda_i$, and its updated value $\lambda_i'$

$$\nabla A\left(\eta_0 + \lambda_i' + \sum_{j \neq i} \lambda_j\right) = -\beta_i \nabla_{\lambda_i} A_i((\theta - \beta_i^{-1}\lambda_i, \beta_i^{-1}))$$
$$= \mathbb{E}_{p_i(z)}[s(z)] \qquad (36)$$

where $p_i \in \mathcal{F}_i$ denotes the $i$-th *tilted distribution*, defined by (6). Applying $\nabla A^*(.)$, the inverse of $\nabla A(.)$, to both sides of (36), and rearranging terms, we recover the moment-matching update of EP

$$\lambda_i \leftarrow \nabla A^*\left(\mathbb{E}_{p_i(z)}[s(z)]\right) - \eta_0 - \sum_{j \neq i} \lambda_j. \qquad (37)$$

Often a damping parameter is used to aid convergence, giving the more general update

$$\lambda_i \leftarrow (1 - \alpha)\lambda_i + \alpha\left(\nabla A^*\left(\mathbb{E}_{p_i(z)}[s(z)]\right) - \eta_0 - \sum_{j \neq i} \lambda_j\right)$$
$$\leftarrow \lambda_i + \alpha\left(\nabla A^*\left(\mathbb{E}_{p_i(z)}[s(z)]\right) - \eta_0 - \sum_j \lambda_j\right), \qquad (38)$$

where the level of damping is given by $(1 - \alpha)$. It was shown by Heskes and Zoeter [25] that (4) follows a decrease direction in $L$ with respect to $\lambda_i$, and so is guaranteed to decrease $L$ when $\alpha$ is small enough.

## D Natural gradient view of EP

Let $\tilde{p}_i^{(t)}$ be the member of $\mathcal{F}$ with natural parameter $\eta_i^{(t)}(\lambda_i) = \eta_0 + \sum_j \lambda_j^{(t)} + \alpha^{-1}(\lambda_i - \lambda_i^{(t)})$. Proposition 1 states that that the moment-matching updates of EP can be viewed as performing NGD in $L$ with respect to the mean parameters, $\mu_i$, of $\tilde{p}_i^{(t)}$. We restate the proposition below, and give a proof. Note that the right hand side of (7) maps the EP update (4) onto mean parameters of $\tilde{p}_i^{(t)}$. The left hand side of (7) is simply a NGD update in $\mu_i$.

**Proposition 1.** *For $\alpha > 0$, the moment-matching update of EP (4) is equivalent to performing an NGD step in L with respect to the mean parameters of $\tilde{p}_i^{(t)}$ with step size $\alpha^{-1}$. That is, for $\mu_i = \mathbb{E}_{\tilde{p}_i^{(t)}(z)}[s(z)]$, and $\tilde{F}_i^{(t)}(\mu_i)$ the FIM of $\tilde{p}_i^{(t)}$ with respect to $\mu_i$, we have*

$$\mu_i - \alpha^{-1}\left[\tilde{F}_i^{(t)}(\mu_i)\right]^{-1}\frac{\partial L}{\partial \mu_i} = \left(\nabla A \circ \eta_i^{(t)}\right)\left(\lambda_i - \alpha\left(\eta_0 + \sum_j \lambda_j - \nabla A^*\left(\mathbb{E}_{p_i(z)}[s(z)]\right)\right)\right). \quad (7)$$

*Proof.* Taking the left hand side of (7), we have

$$\mu_i - \alpha^{-1}\left[\tilde{F}_i^{(t)}(\mu_i)\right]^{-1}\frac{\partial L_i}{\partial \mu_i} = \mu_i - \alpha^{-1}\left[\tilde{F}_i^{(t)}(\mu_i)\right]^{-1}\frac{\partial \lambda_i}{\partial \mu_i}\frac{\partial L}{\partial \lambda_i}$$
$$= \mu_i - \alpha^{-1}\left(\frac{\partial \eta_i}{\partial \mu_i}\right)^{-1}\frac{\partial \eta_i}{\partial \mu_i}\frac{\partial \lambda_i}{\partial \eta_i}\frac{\partial L}{\partial \lambda_i}$$
$$= \mu_i - \frac{\partial L}{\partial \lambda_i}$$
$$= \mu_i - \left(\nabla A\left(\eta_0 + \sum_j \lambda_j\right) - \mathbb{E}_{p_i(z)}[s(z)]\right)$$
$$= \mu_i - \left(\mu_i - \mathbb{E}_{p_i(z)}[s(z)]\right)$$
$$= \mathbb{E}_{p_i(z)}[s(z)], \qquad (39)$$

where the penultimate equality follows from the definition $\mu_i = \mathbb{E}_{\tilde{p}_i^{(t)}(z)}[s(z)]$, and the fact that $\lambda_j^{(t)} = \lambda_j \, \forall \, j$ before the update. All that remains is to show that the right hand side of (7) is equal

to $\mathbb{E}_{p_i(z)}[s(z)]$.

$$\left(\nabla A \circ \eta_i^{(t)}\right)\left(\lambda_i - \alpha\left(\eta_0 + \sum_j \lambda_j - \nabla A^*\left(\mathbb{E}_{p_i(z)}[s(z)]\right)\right)\right)$$

$$= \nabla A\left(\alpha^{-1}(\lambda_i - \lambda_i^{(t)}) + \nabla A^*\left(\mathbb{E}_{p_i(z)}[s(z)]\right)\right)$$

$$= \mathbb{E}_{p_i(z)}[s(z)] \tag{40}$$

$\square$

## E EP-$\eta$ update direction derivation

Using the relation $\eta_i = \eta_i^{(t)}(\lambda_i)$, and therefore $\lambda_i = (\eta_i^{(t)})^{-1}(\eta_i)$, we have

$$-\left[\tilde{F}_i(\eta_i)\right]^{-1}\frac{\partial L}{\partial \eta_i} = -\left[\tilde{F}_i(\eta_i)\right]^{-1}\frac{\partial \lambda_i}{\partial \eta_i}\frac{\partial L}{\partial \lambda_i}$$

$$= -\alpha\left(\frac{\partial \mu_i}{\partial \eta_i}\tilde{F}_i(\mu_i)\frac{\partial \mu_i}{\partial \eta_i}\right)^{-1}\frac{\partial L}{\partial \lambda_i}$$

$$= -\alpha\left(\frac{\partial \eta_i}{\partial \mu_i}\tilde{F}_i(\mu_i)^{-1}\frac{\partial \eta_i}{\partial \mu_i}\right)\frac{\partial L}{\partial \lambda_i}$$

$$= -\alpha\left(\frac{\partial \eta_i}{\partial \mu_i}\frac{\partial \mu_i}{\partial \eta_i}\frac{\partial \eta_i}{\partial \mu_i}\right)\frac{\partial L}{\partial \lambda_i}$$

$$= -\alpha\frac{\partial \eta_i}{\partial \mu_i}\left(\nabla A\left(\eta_0 + \sum_j \lambda_j\right) - \mathbb{E}_{p_i(z)}[s(z)]\right). \tag{41}$$

## F EP-$\mu$ update derivation

For $\alpha = 1$, we have

$$\mu_i - \epsilon\left[\tilde{F}_i(\mu_i)\right]^{-1}\frac{\partial L}{\partial \mu_i} = \mu_i - \epsilon\left[\tilde{F}_i(\mu_i)\right]^{-1}\frac{\partial \eta_i}{\partial \mu_i}\frac{\partial \lambda_i}{\partial \eta_i}\frac{\partial L}{\partial \lambda_i}$$

$$= \mu_i - \epsilon\left(\frac{\partial \eta_i}{\partial \mu_i}\right)^{-1}\frac{\partial \eta_i}{\partial \mu_i}\frac{\partial \lambda_i}{\partial \eta_i}\left(\nabla A\left(\eta_0 + \sum_j \lambda_j\right) - \mathbb{E}_{p_i(z)}[s(z)]\right)$$

$$= \mu_i - \epsilon\left(\nabla A\left(\eta_0 + \sum_j \lambda_j\right) - \mathbb{E}_{p_i(z)}[s(z)]\right). \tag{42}$$

## G Bias of EP and EP-$\mu$

Let the bias in dimension $d$ of $\mu_i^{(t+1)}$, after an update at time $t$, be defined as $\mathbb{E}[\mu_i^{(t+1)} - \bar{\mu}_i^{(t+1)}]_d$, where $\bar{\mu}_i^{(t+1)}$ is the value of $\mu_i^{(t+1)}$ after a noise-free update, and expectation is taken over the sampling distributions of all parallel updates at time $t$. Propositions 2 and 3, which we restate and prove below, summarise the effect of decreasing $\alpha$ and $\epsilon$ on EP and EP-$\mu$, respectively.

**Proposition 2.** *After update (4) is executed in parallel over $i$, as $\alpha \to 0^+$, both the expected decrease in $L$, and the bias $\mathbb{E}[\mu_i^{(t+1)} - \bar{\mu}_i^{(t+1)}]_d$, are $O(\alpha)$ for all $d$.*

*Proof.* Let $\eta_i^{(t)}$ and $\mu_i^{(t)}$ be the pre-update natural and mean parameters of $\tilde{p}_i^{(t)}$ respectively. Furthermore, let

$$\mu_i'^{(t)} = \mu_i^{(t)} - \alpha^{-1}\left[\tilde{F}_i^{(t)}(\mu_i)\right]^{-1}\frac{\partial \lambda_i}{\partial \mu_i}\frac{\partial L}{\partial \lambda_i} + \xi_i$$

$$= \mathbb{E}_{p_i(z)}[s(z)] + \xi_i, \tag{43}$$

be the mean parameters of $\tilde{p}_i^{(t)}$ after an EP inner update (4) but *before* being mapped to mean parameters of $\tilde{p}_i^{(t+1)}$ through map (9). $\xi_i$ is some zero-mean noise, e.g. due to MC variation. The

second equality follows from the results of Appendix D. Similarly, let $\eta_i'^{(t)} = \nabla A^*(\mu_i'^{(t)})$ be the corresponding natural parameters. Now apply map (9) to convert to mean parameters of $\tilde{p}_i^{(t+1)}$,

$$\mu_i^{(t+1)} = \left(\nabla A \circ \eta_i^{(t+1)} \circ (\eta_i^{(t)})^{-1} \circ \nabla A^*\right)(\mu_i'^{(t)})$$

$$= \nabla A\left(\left(\eta_i^{(t+1)} \circ (\eta_i^{(t)})^{-1}\right)\left(\nabla A^*(\mu_i'^{(t)})\right)\right)$$

$$= \nabla A\left(\nabla A^*(\mu_i^{(t)}) + \alpha \sum_j \left(\nabla A^*(\mu_j^{(t)'}) - \nabla A^*(\mu_j^{(t)})\right)\right), \quad (44)$$

where the last equality follows from the definition of $\eta_i^{(t)}(.)$, and by observing that

$$\lambda_j^{(t+1)} - \lambda_j^{(t)} = \alpha(\eta_j' - \eta_j)$$

$$= \alpha\left(A^*(\mu_j^{(t)'}) - \nabla A^*(\mu_j^{(t)})\right), \quad (45)$$

for all $j$. Let $\bar{\mu}_i'^{(t)}$ be the mean parameters of $\tilde{p}_i^{(t)}$ after a *noise-free* update, but before mapping to mean parameters of $\tilde{p}_i^{(t+1)}$, so that $\mu_i'^{(t)} = \bar{\mu}_i'^{(t)} + \xi_i$. Substituting this in, we have

$$\mu_i^{(t+1)} = \nabla A\left(\nabla A^*(\mu_i^{(t)}) + \alpha \sum_j \left(\nabla A^*(\bar{\mu}_j^{(t)'} + \xi_j) - \nabla A^*(\mu_j^{(t)})\right)\right). \quad (46)$$

Taking a Taylor expansion around $\alpha = 0$, we have

$$\mu_i^{(t+1)} = \mu_i^{(t)} + \alpha \left[\nabla^2 A\left(\nabla A^*(\mu_i^{(t)})\right)\right]\left(\sum_j \nabla A^*(\bar{\mu}_j^{(t)'} + \xi_j) - \nabla A^*(\mu_j^{(t)})\right) + O(\alpha^2). \quad (47)$$

Now subtract the noise-free update $\bar{\mu}_i^{(t+1)}$, and take expectations with respect to $\xi_j$ for $j = 1, \ldots, m$,

$$\mathbb{E}\left[\mu_i^{(t+1)} - \bar{\mu}_i^{(t+1)}\right] = \mathbb{E}\left[\alpha \left[\nabla^2 A\left(\nabla A^*(\mu_i^{(t)})\right)\right]\left(\right.\right.$$

$$\left.\left.\sum_j \nabla A^*(\bar{\mu}_j^{(t)'} + \xi_j) - \nabla A^*(\bar{\mu}_j^{(t)'})\right) + O(\alpha^2)\right]. \quad (48)$$

the first term on the right hand side does not have zero expectation in general—$\nabla A^*(.)$ is not generally affine—and so the bias is $O(\alpha)$ as $\alpha \to 0^+$.

To see that the expected change in $L$ is also $O(\alpha)$, take the Taylor expansion of $L$ along the update direction—obtained by subtracting $\lambda_i$ from the right hand side of (4))—as a function of the step size around $\alpha = 0$,

$$-\alpha\left(\eta_0 + \sum_{j \neq i} \lambda_j - \nabla A\left(E_{p_i(z)}[s(z)] + \xi_i\right)\right)^\top \frac{\partial L}{\partial \lambda_i} + O(\alpha^2), \quad (49)$$

which is clearly $O(\alpha)$ as $\alpha \to 0^+$. $\qquad\square$

**Proposition 3.** *After update (12) is executed in parallel over $i$, as $\epsilon \to 0^+$, the expected decrease in $L$ is $O(\epsilon)$, and the bias $\mathbb{E}\left[\mu_i^{(t+1)} - \bar{\mu}_i^{(t+1)}\right]_d$ is $O(\epsilon^2)$, for all $d$.*

*Proof.* Let $\eta_i^{(t)}$ and $\mu_i^{(t)}$ be the pre-update natural and mean parameters of $\tilde{p}_i^{(t)}$ respectively. Furthermore, let

$$\mu_i'^{(t)} = \mu_i^{(t)} - \epsilon\left(\nabla A\left(\eta_0 + \sum_j \lambda_j\right) - \mathbb{E}_{p_i(z)}[s(z)] - \xi_i\right)$$

$$= \mu_i^{(t)} - \epsilon\left(\mu_i^{(t)} - \mathbb{E}_{p_i(z)}[s(z)] - \xi_i\right), \quad (50)$$

be the mean parameters of $\tilde{p}_i^{(t)}$ after an EP-$\mu$ update (12) but *before* being mapped to mean parameters of $\tilde{p}_i^{(t)}$ through (9). $\xi_i$ is some zero-mean noise, e.g. due to MC variation. The second equality

follows from the definition $\mu_i^{(t)} = \mathbb{E}_{\tilde{p}_i^{(t)}(z)}[s(z)]$, and the fact that $\tilde{p}_i^{(t)} = p$ before an update. Now apply map (9) to convert to mean parameters of $\tilde{p}_i^{(t+1)}$,

$$
\begin{aligned}
\mu_i^{(t+1)} &= \left(\nabla A \circ \eta_i^{(t+1)} \circ \left(\eta_i^{(t)}\right)^{-1} \circ \nabla A^*\right)\left(\mu_i'^{(t)}\right) \\
&= \nabla A\left(\left(\left(\eta_i^{(t+1)} \circ \left(\eta_i^{(t)}\right)^{-1}\right)\left(\nabla A^*\left(\mu_i'^{(t)}\right)\right)\right)\right) \\
&= \nabla A\left(\nabla A^*\left(\mu_i^{(t)}\right) + \sum_j\left(\nabla A^*\left(\mu_j^{(t)'}\right) - \nabla A^*\left(\mu_j^{(t)}\right)\right)\right),
\end{aligned}
\tag{51}
$$

where the last equality is obtained from the definition of $\eta_i^{(t)}(.)$ for $\alpha = 1$, and using

$$
\begin{aligned}
\lambda_j^{(t+1)} - \lambda_j^{(t)} &= \eta_j' - \eta_j \\
&= A^*\left(\mu_j^{(t)'}\right) - \nabla A^*\left(\mu_j^{(t)}\right)
\end{aligned}
\tag{52}
$$

for all $j$. Substituting in the definition of $\mu_j'^{(t)}$, we have

$$
\mu_i^{(t+1)} = \nabla A\left(\nabla A^*\left(\mu_i^{(t)}\right) + \sum_j\left[\phantom{\Big[}\right.\right.
$$
$$
\left.\left.\nabla A^*\left((1-\epsilon)\mu_j^{(t)} + \epsilon\left(\mathbb{E}_{p_j(z)}[s(z)] + \xi_j\right)\right) - \nabla A^*\left(\mu_j^{(t)}\right)\right]\right).
\tag{53}
$$

Taking a Taylor expansion around $\epsilon = 0$, we have

$$
\begin{aligned}
\mu_i^{(t+1)} &= \mu_i^{(t)} + \epsilon\left[\nabla^2 A\left(\nabla A^*\left(\mu_i^{(t)}\right)\right)\right]\sum_j\frac{\partial}{\partial\epsilon}\Big\{ \\
&\quad \left.\nabla A^*\left((1-\epsilon)\mu_j^{(t)} + \epsilon\left(\mathbb{E}_{p_j(z)}[s(z)] + \xi_j\right)\right)\right\}\Big|_0 + O(\epsilon^2) \\
&= \mu_i^{(t)} + \epsilon\left[\nabla^2 A\left(\nabla A^*\left(\mu_i^{(t)}\right)\right)\right]\left[\nabla^2 A^*\left(\mu_i^{(t)}\right)\right]\sum_j\frac{\partial}{\partial\epsilon}\Big\{ \\
&\quad \left.(1-\epsilon)\mu_j^{(t)} + \epsilon\left(\mathbb{E}_{p_j(z)}[s(z)] + \xi_j\right)\right\}\Big|_0 + O(\epsilon^2) \\
&= \mu_i^{(t)} + \epsilon\left[\nabla^2 A\left(\nabla A^*\left(\mu_i^{(t)}\right)\right)\right]\left[\nabla^2 A^*\left(\mu_i^{(t)}\right)\right]\sum_j\Big( \\
&\quad \mathbb{E}_{p_j(z)}[s(z)] + \xi_j - \mu_j^{(t)}\Big) + O(\epsilon^2).
\end{aligned}
\tag{54}
$$

Finally, subtract the noise free $\bar{\mu}_i^{(t+1)}$, and take expectations (over $\xi_j$ for $j = 1, \ldots, m$) to obtain the bias

$$
\mathbb{E}\left[\mu_i^{(t+1)} - \bar{\mu}_i^{(t+1)}\right] = \mathbb{E}\left[\epsilon\left[\nabla^2 A\left(\nabla A^*\left(\mu_i^{(t)}\right)\right)\right]\left[\nabla^2 A^*\left(\mu_i^{(t)}\right)\right]\sum_j\xi_j + O(\epsilon^2)\right].
\tag{55}
$$

By assumption $E[\xi_j] = \mathbf{0} \; \forall \; j$, hence the first order term disappears, and the bias is $O(\epsilon^2)$.

To see that the expected decrease in $L$ is also $O(\epsilon)$, note that we are simply following the gradient in $L$ with respect to $\mu$, multiplied by the inverse Fisher and scaled by $\epsilon$. Neither the reparameterisation (from $\lambda_i$ to $\mu_i$) or the Fisher depend on $\epsilon$, and so the change in $L$ must be $O(\epsilon)$ as $\epsilon \to 0^+$ by simple Taylor expansion arguments. $\qquad\square$

## H   Implicit geometries of EP variants

In this paper we have shown that the updates of EP, EP-$\eta$, and EP-$\mu$ can be interpreted as performing NGD of $L$ with respect to the distributions $\{p_i^{(t)}\}_i$. The SNEP method of Hasenclever et al. [17], in contrast, performs NGD of $L$ with respect to the *site potentials*, treating them as bona fide distributions in $\mathcal{F}$ for the purposes of NGD – that is, NGD is performed with respect to the distributions with densities given by $\exp(\lambda_i^\top s(z) - A(\lambda_i))$ for $i = 1, \ldots, m$. In Section 3.4, we argued that the statistical manifolds of $\{p_i^{(t)}\}_i$ are far more relevant for the optimisation of $L$ than those of the site potential pseudo-distributions. In this appendix we shall justify this claim.

The loss function $L$, given by (3), has a unique optimum with respect to $\{\lambda_i\}_i$ when $\nabla_{\lambda_i} L(\theta, \lambda_1, \ldots, \lambda_m) = 0 \ \forall \ i$, which implies that

$$\nabla A(\eta_0 + \sum_j \lambda_j) = -\beta_i \nabla_{\lambda_i} A_i(\theta - \beta^{-1}\lambda_i, \beta^{-1}) \Leftrightarrow$$

$$\mathbb{E}_{p(x)}[s(z)] = \mathbb{E}_{p_i(x)}[s(z)]. \tag{56}$$

The minimisation of $L$ with respect to $\{\lambda_i\}_i$ can therefore be viewed as a joint optimisation in the $m+1$ distributions $p, p_1, \ldots, p_m$, with the minimum found when their expected $\mathcal{F}$-statistics match. Intuitively, it seems that optimisation of $L$ with respect to $\lambda_i$ should be performed with consideration for the geometry of $p$ and $p_i$. By instead performing NGD with respect to the distributions $\{p_i^{(t)}\}_i$, in the case of EP, EP-$\eta$ and EP-$\mu$, or with respect to the site potential psuedo-distributions, in the case of SNEP, these methods can be interpreted as using *surrogate* distributions for NGD [46]. It is possible to show that the Fisher information matrix (FIM) of $\tilde{p}_i^{(t)}$ with respect to $\lambda_i$ is equal to that of $p$, up to a scalar constant, and furthermore, it can also be seen as a close approximation to that of $p_i$. This motivates the use of NGD with respect to $\tilde{p}_i^{(t)}$ for the optimisation of $L$. In contrast, the FIMs of the site-potential pseudo-distributions, used by SNEP, can bear little relation to those of $p$ or $p_i$. In particular, the difference will be greatest when the full approximation $\eta_0 + \sum_j \lambda_j$ is significantly different than $\lambda_i$, and we believe this may be why Hasenclever et al. [17] found that increasing the number of sites reduced the convergence speed of SNEP.

We can make these arguments more concrete by considering the curvature of $L$. Taking second derivatives of $L$ with respect to $\lambda_i$, we have

$$\nabla^2_{\lambda_i} L(\theta, \lambda_1, \ldots, \lambda_m) = \nabla^2_{\lambda_i} A(\eta_0 + \sum_j \lambda_j) + \beta_i \nabla^2_{\lambda_i} A_i((\theta - \beta_i^{-1}\lambda_i, \beta_i^{-1})). \tag{57}$$

Let $\lambda_j^{(t)} = \lambda_j$ at time $t$ for $j = 1, \ldots m$, then

$$\nabla^2_{\lambda_i = \lambda_i^{(t)}} L(\theta, \lambda_1, \ldots, \lambda_m) = \nabla^2_{\lambda_i} A(\eta_0 + \sum_j \lambda_j^{(t)} + (\lambda_i - \lambda_i^{(t)})) + \beta_i \nabla^2_{\lambda_i} A_i((\theta - \beta_i^{-1}\lambda_i, \beta_i^{-1}))$$

$$= \frac{\partial \mu_i}{\partial \eta_i} + \beta_i \nabla^2_{\lambda_i} A_i((\theta - \beta_i^{-1}\lambda_i, \beta_i^{-1})), \tag{58}$$

where $\eta_i$ and $\mu_i$ are the natural and mean parameters of $\tilde{p}_i^{(t)}$, respectively. The Hessian of the second term will not generally be available in closed form, or easily invertible. We can however find a convenient approximation. Note that after an outer update, at time $t$, we have $\theta = \eta_0 + \sum_j \lambda_j^{(t)}$. Also we have that $c \exp(\lambda_i^\top s(z)) \approx \exp(\ell_i(z))$, at least near convergence, where $c$ is some unknown scalar constant. Combining these, we have

$$\nabla^2_{\lambda_i} A_i((\theta - \beta_i^{-1}\lambda_i, \beta_i^{-1})) = \nabla^2_{\lambda_i} \left\{ \log \int \exp\left((\theta - \beta^{-1}\lambda_i)^\top s(z) + \beta^{-1}\ell_i(z)\right) \mathrm{d}\nu(z) \right\}$$

$$\approx \nabla^2_{\lambda_i} \left\{ \log \int \exp \Bigl( \right.$$

$$\left. \left(\eta_0 + \sum_j \lambda_j^{(t)} - \beta^{-1}\lambda_i + \beta^{-1}\lambda_i^{(t)}\right)^\top s(z) \Bigr) \mathrm{d}\nu(z) \right\}$$

$$= \nabla^2_{\lambda_i} A\left(\eta_0 + \sum_j \lambda_j^{(t)} + \beta^{-1}(\lambda_i^{(t)} - \lambda_i)\right)$$

$$= \nabla^2_{\lambda_i} A\left(\eta_0 + \sum_j \lambda_j^{(t)} + \beta^{-1}(\lambda_i - \lambda_i^{(t)})\right)$$

$$= \beta^{-2} \frac{\partial \mu_i}{\partial \eta_i}. \tag{59}$$

Combining this with (58), we have

$$\nabla^2_{\lambda_i = \lambda_i^{(t)}} L(\theta, \lambda_1, \ldots, \lambda_m) \approx (1 + \beta_i^{-1}) \frac{\partial \mu_i}{\partial \eta_i}, \tag{60}$$

and if we use this curvature approximation to perform a quasi-Newton step in $L$ with respect to $\lambda_i$ and step size $\tilde{\epsilon}$, we have

$$\lambda_i \leftarrow \lambda_i - \tilde{\epsilon}(1 + \beta_i^{-1})^{-1} \frac{\partial \eta_i}{\partial \mu_i} \nabla_{\lambda_i} L(\theta, \lambda_1, \ldots, \lambda_m)$$

$$= \lambda_i - \tilde{\epsilon}(1 + \beta_i^{-1})^{-1} \frac{\partial \eta_i}{\partial \mu_i} \left( \nabla A(\eta_0 + \sum_j \lambda_j) - \mathbb{E}_{p_i(z)}[s(z)] \right). \tag{61}$$

By letting $\epsilon = \tilde{\epsilon}(1 + \beta_i^{-1})^{-1}$ we have equivalence with the update of EP-$\eta$, given by (11). We therefore have that EP-$\eta$ is equivalent to performing a quasi-Newton step in $L$. Furthermore, the approximate curvature becomes exact at the optimum if $\ell_i(z)$ is an affine function of $s(z)$, e.g. if $\exp(\ell_i(z))$ is an (unnormalised) member of $\mathcal{F}$.

In contrast, let us consider the implicit curvature approximation used by SNEP. Let $\gamma_i$ be mean parameters of the member of $\mathcal{F}$ with natural parameter $\lambda_i$. To perform the inner minimisation of the variational objective, SNEP performs NGD in $L$ with respect to $\gamma_i$. The implicit curvature matrix used by the SNEP update is then the Fisher with respect to $\gamma_i$, which is given by $\frac{\partial \lambda_i}{\partial \gamma_i} = \nabla A^*(\gamma_i)$. Now consider the actual curvature of $L$ with respect to $\gamma_i$,

$$\nabla_{\gamma_i}^2 L(\gamma_i) = \frac{\partial \lambda_i}{\partial \gamma_i} \nabla_{\lambda_i}^2 L(\theta, \lambda_1, \ldots, \lambda_m) \frac{\partial \lambda_i}{\partial \gamma_i} + \sum\nolimits_{k=1}^{d} \left[ \frac{\partial L}{\partial \lambda_i} \right]_k \nabla_{\gamma_i}^2 \lambda_i(\gamma_i), \tag{62}$$

where we have used $\lambda_i(\gamma_i) = \nabla A^*(\gamma_i)$ to denote the backwards map. Near the optimum we have $\frac{\partial L}{\partial \lambda_i} \approx \mathbf{0}$, and so the curvature will be approximately equal to the first term as we approach the optimum. We have already shown that

$$\nabla_{\lambda_i}^2 L(\theta, \lambda_1, \ldots, \lambda_m) \approx (1 - \beta_i^{-1}) \frac{\partial \mu_i}{\partial \eta_i} = (1 - \beta_i^{-1}) \nabla A(\eta_0 + \sum\nolimits_j \lambda_j), \tag{63}$$

and so the curvature with respect to $\gamma_i$ is

$$\nabla_{\gamma_i}^2 L(\gamma_i) \approx (1 - \beta_i^{-1}) \frac{\partial \lambda_i}{\partial \gamma_i} \nabla^2 A(\eta_0 + \sum\nolimits_j \lambda_j) \frac{\partial \lambda_i}{\partial \gamma_i}$$

$$= (1 - \beta_i^{-1}) \frac{\partial \lambda_i}{\partial \gamma_i} \nabla^2 A(\eta_0 + \sum\nolimits_j \lambda_j) \left( \nabla^2 A(\lambda_i) \right)^{-1}. \tag{64}$$

Compare this with the implicit curvature used by SNEP of $\frac{\partial \lambda_i}{\partial \gamma_i}$, and it is clear that the two are proportional only when $\eta_0 + \sum_j \lambda_j = \lambda_i$. This suggests that if there are many sites ($m$ is large), or if the prior is very informative, the curvature matrix implicitly used by SNEP is likely to be significantly different from the true curvature, even near the optimum.

## I  Computational cost analysis

Let $c_{\text{samp}}$ be the cost of drawing a single sample from one of the tilted distributions. Let $c_{\text{fwd}}$ be the cost of converting from natural to mean parameters in the base family $\mathcal{F}$ (the forward mapping). Similarly, let $c_{\text{bwd}}$ be the cost of converting from mean to natural parameters in $\mathcal{F}$ (the backward mapping). Recall that $m$ is the number of sites. We now state the computational complexity of a round of parallel updates in each of the EP variants evaluated in this paper.

**EP**  EP draws $n_{\text{samp}}$ samples from each of the $m$ sites. It then performs $m$ backward mappings, to map from moments back to updated site parameters. The total cost is then $mn_{\text{samp}}c_{\text{samp}} + mc_{\text{bwd}}$.

**EP-$\eta$**  EP-$\eta$ draws $n_{\text{samp}}$ samples from each of the $m$ sites. Each update also involves one forward mapping per iteration. In addition, each iteration also requires $m$ JVPs through the backward mapping $\nabla A^*(.)$. The primals of this JVP are the same for each site, so the linearisation only needs to be performed once per update, costing $\approx 2c_{\text{bwd}}$, but we have $m$ tangents. The overall cost of an EP-$\eta$ iteration is therefore $\approx mn_{\text{samp}}c_{\text{samp}} + c_{\text{fwd}} + (2 + m)c_{\text{bwd}}$.

**EP-$\mu$**  EP-$\mu$ draws $n_{\text{samp}}$ samples from each of the $m$ sites. Each update involves one forward mapping, and $m$ backward mappings per iteration, but notably does *not* require any JVPs. The overall cost of an EP-$\mu$ iteration is therefore $mn_{\text{samp}}c_{\text{samp}} + c_{\text{fwd}} + mc_{\text{bwd}}$.

**SNEP**  SNEP draws $n_{\text{samp}}$ samples from each of the $m$ sites, and performs $m$ backward mappings per iteration. The total cost is then $mn_{\text{samp}}c_{\text{samp}} + mc_{\text{bwd}}$, which is the same as EP.

In this work we largely assume that $c_{\text{samp}}$ is the dominant cost. This is often the case when the sampling is performed using MCMC. For example, in the NUTS sampler with default numpyro settings – as used in our evaluation – each sample can involve evaluating up to 1024 gradients of the tilted distribution log density.

The other costs, $c_{\text{fwd}}$ and $c_{\text{bwd}}$, depend on $\mathcal{F}$. In the case of a MVN family with dense covariance, the foward and backward mappings are $O(d^3)$, for $d$ the dimensionality of $z$. When $d$ is large, $c_{\text{fwd}}$ and $c_{\text{bwd}}$ become more significant, in which case the balance will shift in favour of using more samples per update (and taking larger steps). When $d$ is very large, however, it may be necessary to choose a diagonal covariance base family instead, in which case the mappings can be performed in $O(d)$ time. Also note that when $d$ increases, $c_{\text{samp}}$ will also typically increase at a rate faster than $d$, even with optimal tuning [8]. In our evaluation we used just a single sample per update for EP-$\eta$ and EP-$\mu$, and they significantly outperformed the baselines when measure in both NUTS steps *and* wall-clock-time.

We also note that the per-iteration cost of $c_{\text{fwd}}$ and $c_{\text{bwd}}$ could be reduced by exploiting the results of previous iterations. For example, in the case of a MVN with dense covariance, the cost of the forward and backward mappings are dominated by matrix inversions. This cost could be significantly reduced by using the result from the previous iteration to warm-start a Newton-style iterative inversion routine. This approach was used by Anil et al. [2] to reduce the cost of computing inverse matrix roots as part of a wider deep-learning optimisation scheme. We did not make use of such optimisations in our experiments.

## J    Evaluation details

In this appendix, we give details about the experiments presented in Section 4. To evaluate the performance of the different variants, we monitored KL divergence to an estimate of the optimum, obtained by running EP to convergence with a large number of samples ($n_{\text{samp}} = 10^5$ for the final iterations). We used 500 different hyperparameter settings for each variant, chosen using random search, and repeated each run using 5 different random seeds, which were used to seed the MCMC samplers. Hyperparameter settings that failed in any of the 5 runs were discarded. We used this setup for all experiments in Section 4.

The $x$-axis in Figure 2 of Section 4 corresponds to the number of NUTS steps, or more specifically, the number of NUTS candidate evaluations. This number is hardware and implementation agnostic, and roughly corresponds to the total computational cost incurred up to that point. Wall-clock time results are given in Appendix L.

**Hyperparameter search spaces**    The step size for EP, $\alpha$, was drawn log-uniformly in the range $(10^{-4}, 1)$, and $n_{\text{samp}}$ was drawn log-uniformly between $[d + 2.5, 10000.5)$ and then rounded to the nearest integer, where $d$ is the dimensionality of $z$. Note that the debiasing estimator used by Xu et al. [52] for MVN families is not defined for $n_{\text{samp}} \leq d + 2$. The thinning ratio was drawn uniformly from $\{1, 2, 3, 4\}$. The step size for EP-$\eta$ and EP-$\mu$, $\epsilon$, was drawn log-uniformly in the range $(10^{-5}, 10^{-2})$, and $n_{\text{samp}}$ was fixed to 1. The step size for SNEP was drawn log-uniformly in the range $(10^{-5}, 10^{-2})$, with $n_{\text{samp}}$ and $n_{\text{inner}}$ (independently) drawn log-uniformly in the range $[.5, 10.5)$ and then rounded to the nearest integer.

**Double-loops**    All the methods considered can be used in a double-loop manner by taking $n_{\text{inner}} > 1$ inner steps per outer update. We did not find this necessary in any of our experiments, and so we fixed $n_{\text{inner}} = 1$ for EP, EP-$\eta$ and EP-$\mu$. For SNEP we included $n_{\text{inner}}$ in the search space (as described above) to ensure that setting it to 1 was not negatively impacting its performance.

**Hardware**    All experiments were executed on 76-core Dell PowerEdge C6520 servers, with 256GiB RAM, and dual Intel Xeon Platinum 8368Q (Ice Lake) 2.60GHz processors. Each individual run was assigned to a single core.

**Software**    Implementations were written in JAX [7], with NUTS [27] used as the underlying sampler. We used the numpyro [42] implementation of NUTS with default settings. For experiments with a NIW base family, we performed mean-to-natural parameter conversions using the method of So [45], with JAXopt [5] used to perform implicit differentiation through the iterative solve.

**MCMC hyperparameter adaptation**    We performed regular warm-up phases in order to adapt the samplers to the constantly evolving tilted distributions, consistent with prior work [52, 48]. The frequency and duration of these warm-up phases was also included in the hyperparameter search as follows. We drew the duration (number of samples) of each warm-up phase log-uniformly in the

range $[99.5, 1000.5)$ and then rounded to the nearest integer. We then drew the sampling-to-warm-up ratio log-uniformly in the range $(1, 4)$. This ratio then determined how frequently the warm-up was performed, which we rounded to the nearest positive integer number of updates. Note that the Pareto frontier plots in Figures 2 and 7 include the computation / time spent during warm-up phases.

**Site parameter initialisation**  For EP, EP-$\eta$ and EP-$\mu$, we initialised the site parameters to $\lambda_i = \mathbf{0} \; \forall \, i$ for all models, with the exception of the cosmic radiation model for reasons we discuss later. SNEP is not compatible with improper site potentials, and so for models where $\mathcal{F}$ was MVN we used $\lambda_i = (2m)^{-1}\eta_0 \; \forall \, i$, consistent with Vehtari et al. [48]. Unfortunately this initialisation strategy for SNEP is not always valid when $\mathcal{F}$ is NIW. This is because the site potentials in SNEP are restricted to being proper distributions, and this constraint was violated when using this initialisation in our experiments. We tried various other initialisation strategies, but none produced satisfactory results or allowed us to make a meaningful comparison, hence we omitted SNEP from the experiments with NIW $\mathcal{F}$; namely, the political survey hierarchical logistic regression and neural response model experiments.

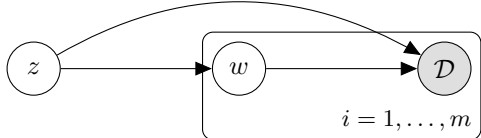

Figure 4: Directed graphical model for the experiments of Section 4.

We set $\beta_i = 1 \; \forall \, i$ in all experiments, corresponding to the original (not power) EP of Minka [37]. All of the models in these experiments followed the same general structure, given by

$$p_0(z) \prod_i p(w_i \mid z) p(\mathcal{D}_i \mid w_i, z), \tag{65}$$

where $p_0$ is a member of a tractable, minimal exponential family $\mathcal{F}$. This is shown graphically in Figure 4. We now provide details about individual experiments.

## J.1 Hierarchical logistic regression

In the hierarchical logistic regression (HLR) experiments, there were $m$ groups (logistic regression problems) each with $n$ covariate/response pairs, so that $\mathcal{D}_i = \{(x_{i,j}, y_{i,j})\}$, $x_{i,j} \in \mathbb{R}^d$, and $y_{i,j} \in \{0, 1\}$, for $i = 1$ to $m$, and $j = 1$ to $n$. Each group had its own unobserved vector of regression coefficients $w_i \in \mathbb{R}^d$, with density $p(w_i \mid z)$ parameterised by global parameters $z$.

**Synthetic data with MVN prior**  In the synthetic experiment, we had $m = 16$ groups, with $d = 4$ and $n = 20$. $\mathcal{F}$ was an MVN family, and $z = (\mu_1, \log \sigma_1^2, ..., \mu_4, \log \sigma_4^2) \in \mathbb{R}^8$ corresponded to the means and log-variances for the independent normal density $p(w_i \mid z) = \prod_j \mathcal{N}(w_{i,j} \mid \mu_j, \sigma_j^2)$.[8] The prior on $z$ had density $p_0(z) = \mathcal{N}(\mathbf{0}, \mathrm{diag}((4, 2, 4, 2, \ldots)))$, where $\mathrm{diag}(.)$ constructs a diagonal matrix from its vector-valued argument. The dataset was generated using the same procedure as Vehtari et al. [48].

**Political survey data with NIW prior**  In the political survey data experiment, $\mathcal{F}$ was a normal-inverse-Wishart (NIW) family of distributions, and $z = (\mu, \mathrm{vech}(\Sigma))$ was used to parameterise the group-level coefficient density $p(w_i \mid z) = \mathcal{N}(w_i \mid \mu, \Sigma)$. $p_0$ was a NIW prior on $\mu$ and $\Sigma$, with parameters $\mu_0 = \mathbf{0}$, $\nu = 9$, $\lambda = 1$, and $\Psi = I$ (following Wikipedia notation). We used a log-Cholesky parameterisation of $\Sigma$ for the purposes of sampling. The dataset consisted of binary responses to the statement *"Allow employers to decline coverage of abortions in insurance plans (Support / Oppose)"*. We constructed $m = 50$ regression problems, corresponding to the 50 US states, and truncated the data so that there were exactly $n = 97$ responses for each state. We used 6 predictor variables from the dataset, corresponding to characteristics of a given survey participant. The predictors were binary variables conveying: age (3 groups), ethicity (2 groups), education (3

---

[8]Note that in this section we use $\mu$ to denote the mean of a normal or multivariate normal distribution. This should not to be confused with the usage for exponential family mean parameters in the main text.

groups) and gender. We also included a state-level intercept to capture variation in the base response level between states, so that $d = 7$. This setup is based on one used by Lopez-Martin et al. [33]. The data is available at `https://github.com/JuanLopezMartin/MRPCaseStudy`.

## J.2 Cosmic radiation model

In this experiment, there were $m$ nonlinear regression problems, each corresponding to a model of the relationship between diffuse galactic far ultraviolet radiation (FUV) and 100-μm infrared (i100) emission in a particular sector of the observable universe.

The nonlinear regression model for sector $i$ had parameters $w_i \in \mathbb{R}^9$. The $m$ regression problems were related through common hyperparameters $z \in \mathbb{R}^{18}$, which parameterised the section-level densities $p(\mathcal{D}_i, w_i \mid z)$ where $\mathcal{D}_i$ is the observed data for sector $i$. $\mathcal{F}$ was the family of MVN distributions, and the prior on $z \in \mathbb{R}^{18}$ had density $p_0(z) = \mathcal{N}(\mathbf{0}, 10I)$. The specifics of $p(\mathcal{D}_i, w_i \mid z)$ are quite involved and we refer the reader to Vehtari et al. [48] for further details.

Vehtari et al. [48] applied this model to data obtained from the Galaxy Evolution Explorer telescope. We were unable to obtain the dataset, and so we generated synthetic data using hyperparameters that were tuned by hand to try and match the qualitative properties of the original data set (see Appendix N for examples). We used a reduced number of $m = 36$ sites and $n = 200$ observations per site to reduce computation, allowing us to perform a comprehensive hyperparameter search.

Using the conventional site parameter initialisation of $\lambda_i = \mathbf{0} \; \forall \; i$ resulted in most runs of EP failing during early iterations. We found that this was resolved by initialising with the method used by Vehtari et al. [48] for SNEP, that is, $\lambda_i = (2m)^{-1}\eta_0 \; \forall \; i$, and so we used this initialisation for all methods. We note, however, that the performances of EP-$\eta$ and EP-$\mu$ were largely unaffected by this change.

## J.3 Neural response model

In this experiment we performed inference in a hierarchical Bayesian neural response model, using recordings of V1 complex cells in an anaesthetised adult cat. 10 neurons in a specific area of cat V1 were simultaneously recorded under the presentation of 18 different visual stimuli, each repeated 8 times, for a total of 144 trials. Neural data were recorded by Tim Blanche in the laboratory of Nicholas Swindale, University of British Columbia, and downloaded from the NSF-funded CRCNS Data Sharing website `https://crcns.org/data-sets/vc/pvc-3` [4].

$z = (\mu, \mathrm{vech}(\Sigma)) \in \mathbb{R}^{65}$ was used to parameterise $\mathcal{N}(\log r_j; \mu, \Sigma)$, the density for latent log firing rates in trial $j$, for $j = 1, \ldots, 144$. $p_0 \in \mathcal{F}$ was NIW with parameters $\mu_0 = \mathbf{1}$, $\nu = 12$, $\lambda = 2.5$, and $\Psi = 1.25I$ (again following Wikipedia notation). The observed spike count for neuron $k$ in trial $j$, $x_{j,k} \in \mathbb{N}$, was modelled as $\mathrm{Poisson}(c_{j,k}, \exp(r_{j,k}))$, for $j = 1, \ldots, 144$, $k = 1, \ldots, 10$. We again used a log-Cholesky parameterisation of $\Sigma$ for sampling.

We grouped trials together into $m = 8$ batches of $n = 18$ trials, so that $w_i = (\log r_{180(i-1)+1}, \ldots, \log r_{180i}) \in \mathbb{R}^{180}$ was the concatenation of log firing rates for all 10 neurons across 18 trials, with $\mathcal{D}_i = \{(x_{j,1}, \ldots, x_{j,10})\}$ the observed spike counts, for $j = 180(i-1)+1$ to $180i$.

## K  Hyperparameter sensitivity

In this appendix, we examine the effect of varying hyperparameters of EP and EP-$\eta$ on the synthetic hierarchical logistic regression experiment of Section 4. The results for EP-$\mu$ are similar to those of EP-$\eta$ and so we do not consider it separately here.

**EP**   In the left panel of Figure 5 we show the effect of varying the number of samples used for estimating updates in EP – more accurate regions of the frontier generally require more samples. The middle panel, similarly, shows the effect of varying the step size, and the right panel shows the effect of varying the thinning ratio $\tau$. Together these plots illustrate the difficulty of tuning EP in stochastic settings – tracing out the frontier is a three-dimensional problem. For example, to achieve the best accuracy within a compute budget of $10^7$ steps, the practitioner would need to set

$400 < n_{\mathrm{samp}} \le 500$, $0.1 < \alpha \le 0.3$, and $\tau = 2$, and any deviation from this would seemingly result in suboptimal accuracy.

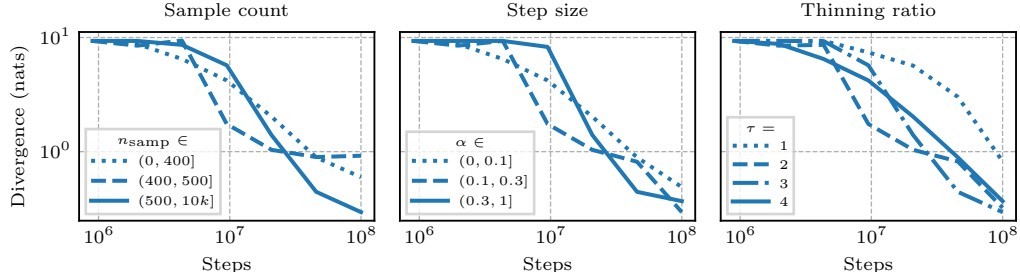

Figure 5: The effect of varying EP hyperparameters. Partial Pareto frontiers show the number of NUTS steps ($x$-axis) against the KL divergence from $p$ to an estimate of the optimum ($y$-axis).

**EP-$\eta$**   In the left panel of Figure 6 we show the effect of varying the number of samples used for estimating updates in EP-$\eta$ – the frontier is traced out with $n_{\mathrm{samp}} = 1$. We note however that the difference between $n_{\mathrm{samp}} = 1$ and $n_{\mathrm{samp}} = 10$ is relatively small, and so it may be sensible to choose $n_{\mathrm{samp}} > 1$ to make efficient use of parallel hardware, or to minimise per-iteration overheads (see Appendix I). The middle panel, similarly, shows the effect of varying the step size, with more accurate regions of the frontier corresponding to a smaller step size. This also suggests that in practice it may make sense to set $\epsilon$ relatively large at first and then gradually decay it to improve the accuracy. Finally, the right panel shows the effect of varying $n_{\mathrm{inner}}$, the number of inner steps performed per outer update. $n_{\mathrm{inner}} > 1$ corresponds to using EP-$\eta$ in "double-loop" mode. We did not find it necessary to increase $n_{\mathrm{inner}}$ above one to obtain convergence in our experiments, but Figure 6 (right) demonstrates that doing so would have a relatively small impact on performance. Together these plots illustrate that hyperparameter tuning for EP-$\eta$ is relatively straightforward. The frontier can be largely be traced out by varying $\epsilon$, and is relatively insensitive to $n_{\mathrm{samp}}$ and $n_{\mathrm{inner}}$.

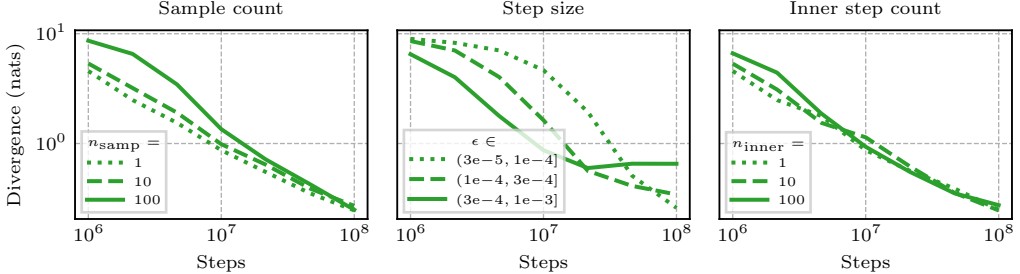

Figure 6: The effect of varying EP-$\eta$ hyperparameters. Partial Pareto frontiers show the number of NUTS steps ($x$-axis) against the KL divergence from $p$ to an estimate of the optimum ($y$-axis).

## L   Pareto frontiers in wall-clock time

In Figure 7, we show Pareto frontiers, with the y-axis showing the lowest KL divergence achieved by any hyperparameter setting, and the x-axis shows the cumulative number of seconds elapsed – that is, the wall-clock time equivalent of Figure 2. These results are in broad agreement with those of Figure 2, which suggests that the sampling cost does indeed dominate the computational overheads of EP-$\eta$ and EP-$\mu$ in these experiments. The wall-clock time results for EP-$\eta$ and EP-$\mu$ would likely be improved by using more than one sample per update, by making more efficient use of hardware resources and minimising per-iteration overheads.

We note that these times are necessarily implementation and hardware dependent. We did not make particular efforts to optimise for wall-clock time. In Appendix I we discuss approaches for minimising per-iteration overheads. These would likely improve wall-clock time performance for all methods, but should disproportionately favour EP-$\eta$ and EP-$\mu$, due to their frequent updates and larger per-iteration overheads compared to EP and SNEP.

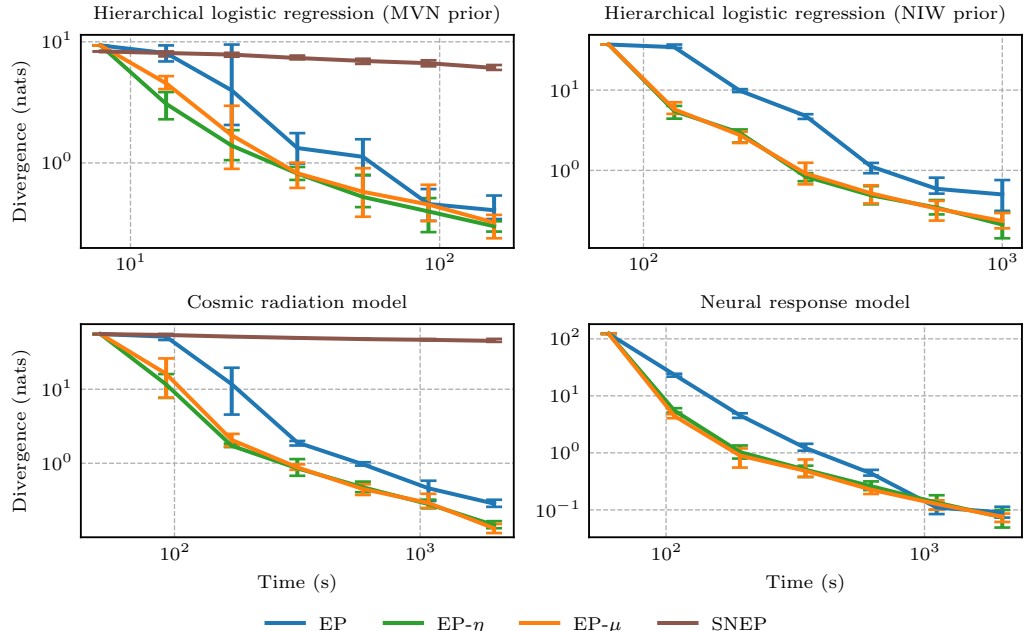

Figure 7: Pareto frontiers showing the number of seconds elapsed ($x$-axis) against the KL divergence from $p$ to an estimate of the optimum ($y$-axis). Each point on the plot marks the lowest KL divergence attained by *any* hyperparameter setting by that time. Error bars mark the full range of values for the marked hyperparameter setting across 5 random seeds.

## M  Comparison with conjugate-computation variational inference (CVI)

Our focus in this work has been on developing improved methods for performing EP when the updates must be estimated with noise. A major motivation for this setting is that it potentially enables EP to be used as a so-called *black-box* inference method, significantly expanding the set of models it can be applied to. EP has several computational advantages over direct MCMC approaches, the prevailing dominant class of black-box mehods, as discussed by earlier works [52, 17, 48]. There is another popular class of black-box inference methods however; namely, those of *variational inference* (VI).[9] In order to understand the trade-offs involved in using EP and its variants over VI, we performed experiments to explore the relative strengths and weaknesses of EP-$\eta$ and conjugate-computation variational inference (CVI) [31], an efficient VI method.

**Experimental setup**  We used EP-$\eta$ and CVI to perform approximate Bayesian inference using the same hierarchical logistic regression model (with MVN prior) and synthetic dataset as Section 4. For our evaluation metrics, we used both forward and reverse KL divergences between the current (MVN) approximation, and a MVN "target" distribution. The target distribution was a MVN distribution fitted using 500,000 samples from the posterior, obtained by running NUTS for 1 million samples and then discarding the first half as a warm-up. For CVI, we used a structured MVN approximation that had the same conditional independence structure as the true posterior – that is, it modelled all pairwise dependencies within $z$, within each $w_i$, and between $z$ and each $w_i$, but did not model direct dependence between $w_i, w_j$ for $i \neq j$. We used 500 random hyperparameter settings for each method, using the search spaces described below. Unless stated otherwise (below), all other details were as described in Appendix J.

**Hyperparameter search spaces**  For EP-$\eta$ we used the same hyperparameter search space as described in Section J. For CVI, the step size was drawn log-uniformly in the range $(10^{-5}, 1)$, and the number of samples used to estimate the update was drawn log-uniformly in the range $[.5, 100.5)$ and then rounded to the nearest integer. The site parameters for CVI were initialised to be the

---

[9]Expectation propagation is also a variational inference method, but we follow convention here by using variational inference to refer specifically to variational optimisation of the free energy / evidence lower bound.

parameters of a zero-mean MVN distribution with scaled identity covariance matrix, with the scale drawn log-uniformly from $(10^{-5}, 10^5)$.

**Wall-clock time performance comparison**   We compared the wall-clock time performance of EP-$\eta$ and CVI, with the former using NUTS as the underlying sampler. For both methods, we chose the hyperparameters that gave the lowest *reverse* divergence between the approximation and the MCMC target distribution after 100 seconds. The results of this experiment are shown in the left panel of Figure 3. We can see that EP-$\eta$ converges at least one order of magnitude slower than CVI on this problem, when measured in wall-clock time. This difference is largely due to the relative cost of drawing samples for the two methods. While CVI also uses samples to estimate its updates, it uses samples from the current MVN approximation, which are cheap to generate using standard methods. EP-$\eta$ on the other hand (as with other EP variants), requires samples from the *tilted* distributions to estimate the updates, and in this experiment we used NUTS to draw those samples. Each NUTS sample can involve many evaluations of the energy function gradient, and furthermore, because MCMC samples are generally autocorrelated, the cost of drawing approximately *independent* samples can be even higher. We used NUTS to be consistent with prior work, and because our focus in this work has primarily been the relative performance of different EP variants, we did not make particular efforts to improve the performance of the underlying samplers, beyond tuning their hyperparameters in a fairly standard manner. However, all of the EP variants considered in this paper, including EP-$\eta$, are agnostic to the choice of sampling kernel used, and we believe there is scope to significantly improve the efficiency of the underlying samplers, for reasons we briefly discuss in Section 6.

**Sample efficiency**   In order to decouple the performance of EP-$\eta$ from that of the underlying sampler, we repeated the experiment using an "oracle" sampling kernel for EP-$\eta$. This oracle kernel was simply NUTS but with a thinning ratio of 100, so that each sample was an approximately independent sample from the tilted distribution. We then compared EP-$\eta$ and CVI using the same metrics as the previous experiment, but with "time" measured in samples. We chose the hyperparameter setting for each method that achieved the lowest reverse KL divergence after 1,500 samples. When measured on this basis we see that the convergence speeds of CVI and EP-$\eta$ are roughly equivalent, demonstrating that the sample efficiency of the two methods are similar. This should not be too surprising, as CVI is (mathematically, but not algorithmically) equivalent to the limiting case of power EP (as $\beta_i \to \infty$) [11, 51].

**Approximation quality**   Figure 3 demonstrates that EP-$\eta$ is able to achieve a more faithful approximation of the target distribution when measured by either the forward *or* reverse KL. At first it may seem surprising that EP is able to obtain a lower *reverse* KL divergence than CVI, but there are two reasons for this apparent anomaly. First, with EP, we have a MVN approximation for $z$ only, with the approximate posterior over local variables ($w_i$ for $i = 1, \ldots, m$) represented only implicitly with samples. CVI on the other hand necessarily optimises a joint MVN over *all* variables, and the $z$ marginal of the optimal VI posterior over all variables is not the same as the optimal VI posterior over $z$. The second reason is that our target distribution is essentially a moment-matched MVN approximation of the true posterior, which would naturally tend to favour the forward KL divergence that is (approximately) targeted by EP. Nevertheless, a qualitative assessment of the pairwise marginals, as seen in the two rightmost panels of Figure 3, shows that the *true* posterior is more faithfully approximated by EP-$\eta$ than CVI, with the latter significantly underestimating uncertainty in the inter-group variability parameters, $\{\log \sigma_i\}_i$. The full set of pairwise posterior marginals can be seen in Figure 8. Note that the CVI approximation appears more constrained than might be expected based on the marginal plots alone. This is due to the zero-avoiding nature of the reverse KL divergence – non-Gaussianity in one variable can constrain the marginal distributions of others, particularly when there is strong dependence. The local variables, $\{w_i\}$, can have both significant non-Gaussian posterior structure, and strong dependence on $\{\log \sigma_i\}_i$.

# N   Cosmic radiation data

Vehtari et al. [48] used a hierarchical Bayesian model to capture the nonlinear relationship between diffuse galactic far ultraviolet radiation (FUV) and 100-μm infrared emission (i100) in various sectors of the observable universe, using data obtained from the Galaxy Evolution Explorer telescope. We were unable to obtain the dataset, and so we generated synthetic data using hyperparameters that

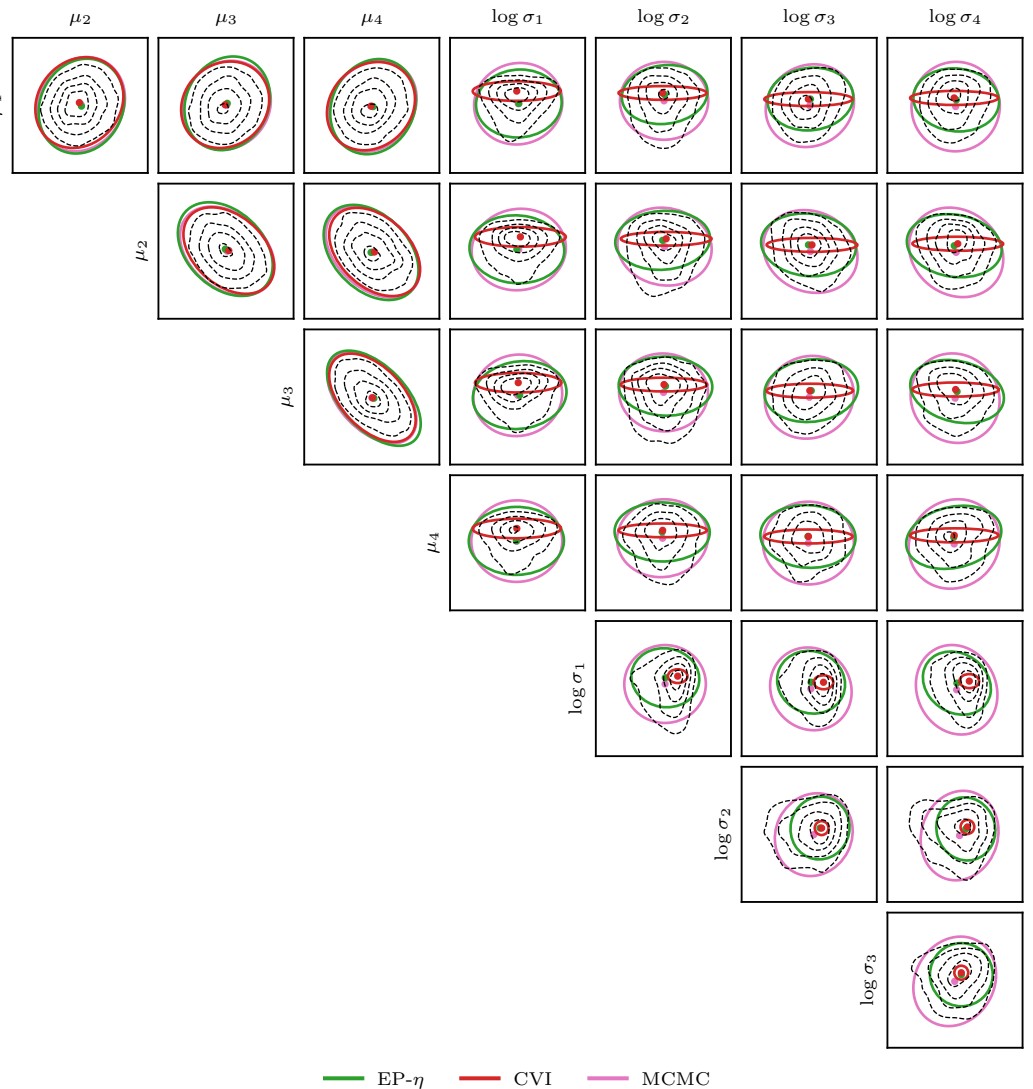

Figure 8: Contours of pairwise posterior marginals for the hierarchical logistic regression experiment of Section 4, overlaid with the MVN approximations of EP-$\eta$, CVI, and one obtained by fitting directly to MCMC samples.

were tuned by hand to try and match the qualitative properties of the original data set. Example data generated using these hyperparameters is shown in Figure 9. For comparison, see Figure 9 of Vehtari et al. [48].

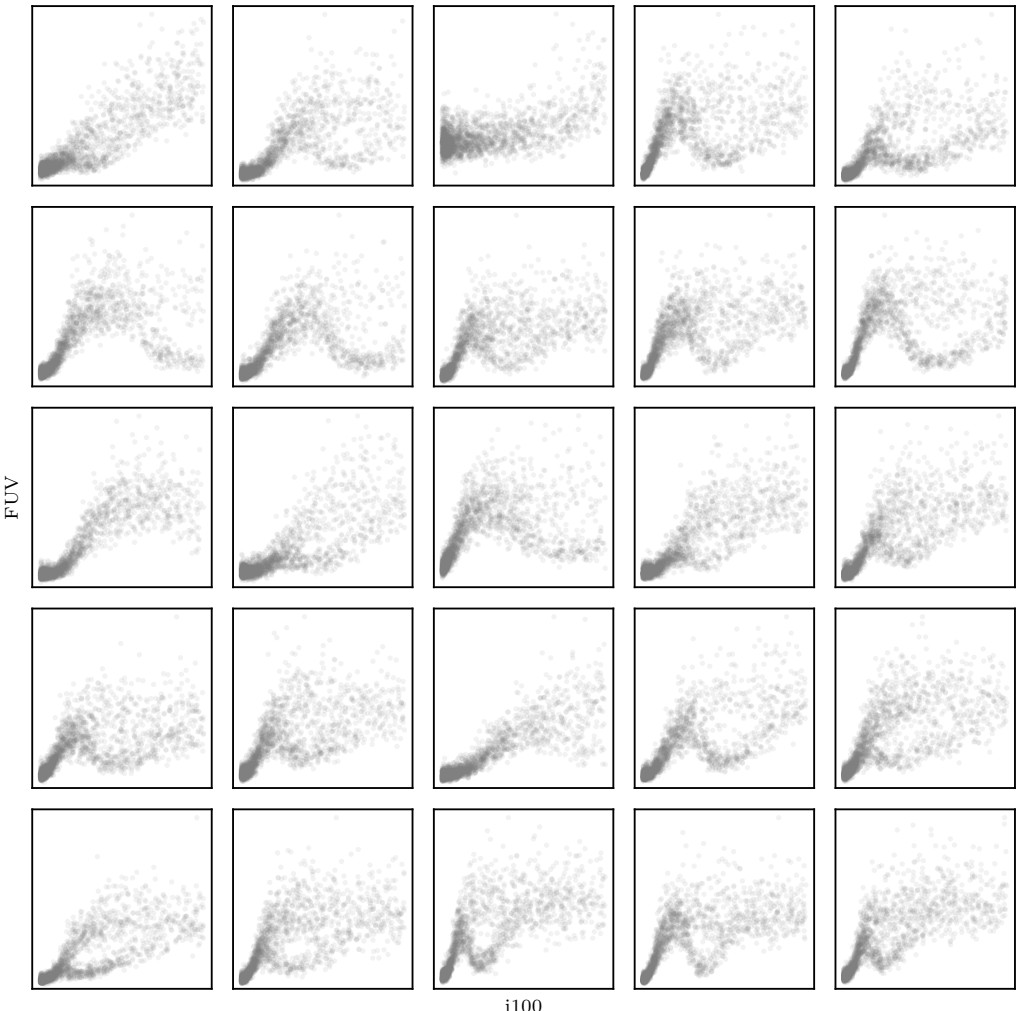

Figure 9: Synthetic data, generated using the cosmic radiation model of Vehtari et al. [48]. Each plot shows galactic far ultraviolet radiation (FUV) versus infrared radiation (i100) for a single sector of the observable universe.

