# OpenReview forum: "Fearless Stochasticity in Expectation Propagation"
_NeurIPS.cc/2024/Conference — NeurIPS 2024 spotlight_

### Official Review · Reviewer_1uyT · 2024-07-01

**Soundness:** 4
**Presentation:** 3
**Contribution:** 3
**Rating:** 6
**Confidence:** 3

**Summary:**

The authors introduce two methods for Expectation Propagation [EP] (which itself can be understood as a method for approximate Bayesian inference) that are robust to the stochastic noise introduced by using approximate expectations in the inner loop of EP. To do so the authors re-interpret the moment-matching update as a particular Natural Gradient Descent (NGD) step. While stochasticity remains due to the expectation, by moving to natural parameters MCMC estimates of the expectation do not suffer from bias-introducing non-linearities. By significantly reducing bias (to zero in the case of EP-$\eta$) the authors show that stable and computationally efficient EP is possible with single-sample moment-matching updates. In experiments the authors show that the method is easier to tune than alternatives and exhibits promising empirical performance.

**Strengths:**

- To my mind the major strength of this submission is that it offers a well-reasoned technical improvement to a class of algorithms (EP) that are arguably underexplored in the literature (probably in large part due to the dominance of variational inference).
- Moreover, empirical results support the effectiveness of the proposed algorithms. Since the authors work hard to reduce dependence on tricky hyperparameter choices, the reader has reason to be confident in the basic validity of the presented empirical results.
- It is conceivable that the (novel, to the best of my knowledge) NGD interpretation offered by the authors could help motivate yet other EP variants.

**Weaknesses:**

To my mind the major weakness of this submission is that the exposition is rather dense and at times hard to follow. This may be somewhat inherent in the technical nature of the topic, but I believe the authors could do a better job of guiding the reader. In particular a lot of space is devoted to somewhat extraneous modeling details ("...between diffuse galactic far ultraviolet radiation and 100-μm infrared emission in various sectors of the observable universe, using data obtained from the Galaxy Evolution Explorer..."), and this space in the main text could be re-purposed to offer more details on the algorithm, discuss the implications of the experimental results in more detail, or give more color on the "unified" EP presentation.

**Questions:**

- math typos: line 804
- NIW is undefined where first introduced
- i guess in sec. 2.1 you should emphasize more that the domain of $\theta_i$ is $\Omega$, since $\theta_i \notin \Theta_i$ as one might expect? similarly i guess in practice given the form of the outer update $\theta_i = \theta_j$ for all $i, j > 0$? maybe this should be emphasized? can you please clarify?
- i understand that this is a paper about EP, but wouldn't it be valuable to benchmark against variational inference in at least one case? whether the authors like it or not, this remains a relevant comparison in my opinion.

**Limitations:**

To my mind the major limitation is the lack of comparison to alternative methods for approximate inference like variational inference.

---

> ### Author Rebuttal · Authors · 2024-08-06
>
> We would like to thank the reviewer for taking the time to read our paper and for providing thoughtful feedback. We agree with the reviewer that the exposition can be dense at times; as the reviewer correctly mentioned, this was to some extent unavoidable due to the nature of the ideas being discussed, but there were several places that we would have liked to expand on points or offer additional explanation and were unable to do so due to space constraints. With the additional page we will look to add additional signposting throughout, including the following specific changes in line with the reviewers suggestions:
> 1. We will add sentences to add some intuition and summarise the idea behind EP-$\eta$ on line 158, before _“We call the resulting procedure…”_, and likewise for EP-$\mu$, on line 184, before _“We call this variant…”_.
> 2. We will expand on the summary of experimental results (lines 251-259) and discuss their implications. We will also add a brief summary of the results for each individual experiment at the end of their descriptions (lines 270, 279, 293 and 304). If space permits we will also look to move some of the hyperparameter sensitivity results from Appendix I into this section.
> 3. We will add a paragraph, beginning on line 79, to inform the reader that we are about to introduce a unified EP algorithm, and explain that the reason for doing so is primarily to show how several variants, including the new ones to be introduced later, are related to one another. We will also add an explanation of Algorithm 1, before line 86.
>
> In response to the reviewer’s questions:
> 1. Yes, we will fix this.
> 2. Agreed, we will define it at first usage.
> 3. Yes, agreed and we will emphasise both points. It is indeed the case that after an outer update $\theta_i = \theta_j$ for all $i, j$. We could therefore obtain an equivalent procedure using just a single variable $\theta$ in the outer optimisation, but we kept the current presentation to be consistent with prior work. We believe this presentation may be a legacy of earlier works on double-loop EP where additional model structure was assumed, in which cases the distributions involved in the optimisation can be over different subsets of variables at each site.
> 4. The main focus of our work was to provide more effective methods for optimising the EP variational problem in the presence of Monte Carlo noise, with the case for using EP in such settings having been been made by prior work (Xu et al. 2014, Hasenclever et al. 2017, Vehtari et al. 2020). That being said, we agree it would be interesting to compare VI with the different EP variants on various performance metrics as a function of computation cost / time, to gain understanding of the tradeoffs involved. Due to time constraints we have not been able to run these experiments for the rebuttal, but we will do so and add the results to the final paper. More specifically, we will compare with natural gradient VI (Khan and Lin, 2017) as suggested by another reviewer.
> As an aside, we note that Bui et al. (2017) studied the effect of the power parameter of power EP ($\beta_i$ in our notation), including the limiting case of VI ($\beta_i \rightarrow \infty$); they found that intermediate values of the power parameter were consistently better across tasks and performance metrics when compared to either the EP or VI extremes. As our methods apply to the general power EP objective, they too can be used to find such intermediate approximations.

---

> > ### Comment · Reviewer_1uyT · 2024-08-08
> >
> > I thank the authors for their careful response. I maintain my score as is and hope the other reviewers will join me in recommending publication.

---

### Official Review · Reviewer_YMg2 · 2024-07-13

**Soundness:** 3
**Presentation:** 3
**Contribution:** 3
**Rating:** 8
**Confidence:** 2

**Summary:**

This work addresses the sensitivity of expectation propagation (EP) to the randomness of Monte Carlo (MC) estimates involved in its update steps. It tackles this issue by recasting the moment matching step in EP as natural gradient descent (NGD) in the mean space of an exponential family distribution. The author identifies that the instability of EP to MC noise is due to the nonlinearity of the mirror map defined by the log partition function of the exponential family. By cleverly moving the NGD from the mean space to the natural parameter space, this issue is bypassed.
The author also studies the influence of the stepsize on the accumulation of MC error, finding that decreasing the stepsize of the NGD updates helps to reduce the bias.

**Strengths:**

- This is a well-written work. It starts with a succinct overview of the expectation propagation (EP) algorithm and summarizes several variants of EP updates in a unified manner (in Algorithm 1). This review also makes the contribution of this work very clear. It provides a sufficient discussion on why EP is sensitive to stochasticity and clearly motivates the design of the methodology. I also appreciate the self-contained review of the exponential family, natural gradient descent, and detailed derivation of EP updates provided in the appendix. Although these materials are standard, they help to make the work more accessible to a general audience.

- The feasibility of performing unbiased NGD updates in the natural space of $\tilde p_i$ (Prop1) is a smart observation, and this simple modification makes the inner loop significantly more robust to the stochasticity in MC est. This simple modification (from mean space to natural parameter space) makes the inner loop significantly more robust to the stochasticity in Monte Carlo estimation. I'm indeed impressed that EP can work with single-sample estimation in the inner loop.

- I also appreciate the rigorous investigation of the effect of $\alpha$ and $\epsilon$ to the bias of estimated mean parmaeter.

- The empirical performance of proposed methods are superior; it outperforms the standard EP by a significant margin (as shown in Fig2).

**Weaknesses:**

The technical side of this work is very strong in my perspective, and I don't observe many weakness on this regard.
However, I think the novelty of this NGD perspective is overclaimed. To my knowledge, the relationship between moment matching in the exponential family and NGD is very well known (e.g., [1][2][3]). Once it is identified that $\tidle p_i$ is in the exponential family, the derivation of NGD (in either mean space Prop1 or natural parameter space) becomes quite straightforward. I hope the authors can elaborate on this point, and include discussion on some relavant literature arolund prop 1.

I believe the real contribution of this work lies in the rigorous understanding of the influence of the stepsize on the accumulation of Monte Carlo (MC) error and the identification of the nonlinearity of the mirror map that leads to this error accumulation, which is already impactful to me.




- [1]: Conjugate-Computation Variational Inference : Converting Variational Inference in Non-Conjugate Models to Inferences in Conjugate Models, 2018
- [2]: MCMC-driven learning, 2024
- [3]: Distributed Bayesian Learning with Stochastic Natural Gradient Expectation Propagation and the Posterior Server

**Questions:**

- How do you estimate the KL divergence in the experiments? Maybe I'm missing something, but the target posterior is only known up to a constant, so unbiased KL est is not feasible?

- I'm curious to see the relative performance of this work to NGVI [1] (specified in the Weakness section).

**Limitations:**

The limitation is well discussed.

---

> ### Author Rebuttal · Authors · 2024-08-06
>
> We would like to thank the reviewer for taking the time to read our paper and for providing thoughtful feedback. With regards to the novelty of our NGD interpretation, we agree that the link between NGD and exponential family moment matching is known, and our intention is not to claim otherwise; we will change the wording around Proposition 1 to make this clearer, with reference to the relevant literature.  We do believe however that the connection of this result with the updates of EP is novel; without the identification of an exponential family distribution $\tilde{p}_i$, parameterised by $\lambda_i$ such that the natural parameters are given by the specific affine function $\eta_i^{(t)}(\lambda_i)$, the connection with NGD does not hold. While this is perhaps a straightforward consequence of the link between NGD and moment matching, we do not believe this connection has been made before and it was a necessary insight for the contributions of the later sections. Indeed the only connection we are aware of having been made between the updates of (power) EP and NGD is in the limiting case (as $\beta_i \rightarrow \infty$) for which the updates of power EP and natural gradient variational inference (NGVI) coincide (Bui et al. 2018, Wilkinson et al. 2021).
>
> In response to the questions:
> 1. The metric displayed in the plots shows the KL divergence from the current approximation ($p$) to (an estimate of) a converged EP solution. The converged solution was found by running EP for a large number of iterations, with the moments estimated using a very large number of samples; we discuss this on lines 247 and 923. We felt this was the most meaningful metric for our experiments, given that we were benchmarking the optimisation performance of different EP variants. Another choice we considered was to show the KL divergence between $p$ and a moment-matched Gaussian, for which the moments are obtained by running long MCMC chains. However, we found this metric to be problematic, as it often does not monotonically decrease during optimisation; for example, converged EP solutions tend to have lower variance than the true posterior (see, for example, Vehtari et al. 2020, Cunningham et al. 2011), and so we often saw a pattern of the KL decreasing as the approximate posterior variance shrinks towards the true posterior variance, and then rising as the approximation variance continues to decrease, before eventually settling at a higher KL than the earlier transient. In contrast, the KL divergence to a converged EP solution was typically monotonic decreasing for stable hyperparameter settings, and had the additional benefit of the attainable lower bound being (approximately) zero.
>
> 2. The main focus of our work was to provide more effective methods for optimising the EP variational problem in the presence of Monte Carlo noise, with the case for using EP in such settings having been been made by prior work (Xu et al. 2014, Hasenclever et al. 2017, Vehtari et al. 2020). That being said, we agree it would be interesting to compare NGVI with the different EP variants on various performance metrics as a function of computation cost / time, to gain understanding of the tradeoffs involved. Due to time constraints we have not been able to run these experiments for the rebuttal, but we will do so and add the results to the final paper. As an aside, we note that Bui et al. (2017) studied the effect of the power parameter of power EP ($\beta_i$ in our notation), including the limiting case of VI ($\beta_i \rightarrow \infty$); they found that intermediate values of the power parameter were consistently better across tasks and performance metrics when compared to either the EP or VI extremes. As our methods apply to the general power EP objective, they too can be used to find such intermediate approximations.

---

> > ### Comment · Reviewer_YMg2 · 2024-08-13
> >
> > I thank the authors for the detailed reponse to address my questions. I'm happy to keep my score.

---

### Official Review · Reviewer_6XAG · 2024-07-15

**Soundness:** 3
**Presentation:** 2
**Contribution:** 3
**Rating:** 5
**Confidence:** 3

**Summary:**

This paper considers new inference algorithms for EP. By framing the moment-matching equations of EP as a natural gradient update of a variational objective, they propose two new algorithms that are better suited for reducing/removing the bias introduced when sampling is required.

**Strengths:**

I like the attempt at generalising and encapsulating the EP literature, and I find the `trick’ of mitigating the sampling inducing bias through a change of parameterisation is clever.

**Weaknesses:**

1)	In general I find the paper quite hard to follow and of course this is not helped by the form of the standard EP equations. For example the main point of the paper is to handle the bias introduced by using sampling to estimate certain quantities within the EP equations. However the source of this sampling is only explicitly mentioned on line 132 in text. It would be much clearer if this was explicitly in Alg 1, and would also make it more convincing that Alg. 1 actually encapsulated many EP style algorithms.

2)	Following above some terms are not defined until later on in the paper. For example the convex conjugate in eqn 4, and on line 100 only defined one page later on line 123.

3)	The paper is very descriptive but does not explain results. For example, the experiments only describe the set up, but all results and any conclusions are pushed into the appendix. This makes it hard to actually assess the contributions.

**Questions:**

1)	What is c_samp?

2)	In Eqn 24 should there be a $beta_i$ scaling term?

3)	In fig 1 a) why do most of the curves have a `u’ shape? Additionally why do the dashed orange/green curves not?

4)	Does sequential update (instead of in parallel) affect the bias?

5)	Why do you fix alpha to 1 and introduce a new step size eps? This looks like the only difference between Eqn 10 and the natural gradient part of 11.

6)	The link between natural gradient steps  and EP updates has been established previously (in Heskes and also in Hasenclever). Why is the view taken in paper a novel perspective compared to these previous works?

7)	How does this work relate to `Bayes-Newton Methods for Approximate Bayesian Inference with PSD Guarantees’  Wilkinson et al, 2021 ? Which consider EP style algorithms with a natural gradient framework.

**Limitations:**

yes

---

> ### Author Rebuttal · Authors · 2024-08-06
>
> We would like to thank the reviewer for taking the time to read our paper and for providing thoughtful feedback. The reviewer’s concerns relate to the paper presentation. We will take specific steps to address these, detailed below. In light of these changes, would the reviewer be willing to reconsider their position on the paper?
> 1. We agree the presentation of EP updates is somewhat atypical. We felt it necessary to provide them in the context of the saddle-point problem, as this perspective is crucial for the later sections. Space constraints meant that providing both this and the conventional view in the main paper was difficult. We think this would be best addressed by referring to a short appendix after acknowledging the atypical presentation on line 69. The appendix would provide the conventional view – motivated as KL-divergence-minimising projections into the approximating family – and show that the resulting updates are equivalent to ours. We will also introduce the sampling source earlier by adding the following at the end of line 72: _“The expectation in (4) is most often computed analytically, or estimated using deterministic numerical methods. In principle it can also be estimated by sampling from $p_i$, however, we will later show that the resulting stochasticity can introduce bias into the procedure.”_. Note that this is also discussed starting on line 91. Regarding Algorithm 1, we will highlight the expectation in the update and add the comment _“stochastic estimation of this expectation can lead to biased updates”_.
> 2. $A^*(.)$ is introduced on line 38, but we will add the following text after that definition: _“$A(.)$ and $A^*(.)$ are convex conjugates of one another.”_.  We will also add reminders throughout to help the reader.
> 3. As the results in Figure 2 were similar we summarised them jointly at the beginning of Section 4 (lines 251-259) before giving an overview of the individual experiments. We will expand on this and give a brief summary of the results for each experiment at the end of their descriptions. If space permits we will also move some of the hyperparameter sensitivity results from Appendix I into this section.
>
> In response to the questions:
> 1. $c_\text{samp}$ is the cost of drawing a sample from one of the tilted distributions. It is defined at the beginning of Appendix G (line 884) and not used outside of that section.
> 2. Yes, we will fix this.
> 3. If the step size is too large the expected progress can decrease. This can simply be due to overshooting the minimum along the step direction. Alternatively, as a larger step size also results in higher variance, it can lead to some steps that are far worse than the expected step location, or even outside of the valid domain. If any steps in our sample went out of the valid domain we could not compute the average, and so the line can end abruptly (above a certain step size).
> 4. Bias results from other site parameters changing between updates, which happens in both sequential and parallel settings. We will explain this on line 140 as other readers are likely to have the same question.
> 5. $\alpha$ plays a similar role to a NGD step size in (10), but it also affects the distribution $\tilde{p}_i$ and hence the map from $\lambda_i$ to $\eta_i$. Note that (10) is an update direction in $\eta_i$ whereas (11) is an update for $\lambda_i$. If we did not fix $\alpha=1$, the resulting coefficient in (11) would be $\alpha^2$. To obtain a more faithful NGD interpretation we fix the distribution and vary the step size, which requires introducing $\epsilon$. We will elaborate on this on line 155.
> 6. We do not believe any prior work has made a direct connection between the updates of EP and NGD, except for the limiting case in which power EP coincides with VI (see answer to 7 below). Hasenclever et al. (2017) derived _new_ updates (different from the standard ones) which performed NGD of the variational objective with respect to a different distribution and parameterisation; in contrast, we show that the standard EP updates can already be viewed as performing NGD. We discuss this from line 210, but will amend the wording to make the distinction clearer. We believe the reviewer may be referring to the extended version of Heskes and Zoeter (2002) in which the authors propose two methods for finding saddle-points of the objective. One performs joint gradient ascent/descent in the parameters, and the other is what could be called “standard” double-loop EP. With respect to the first, the authors say it _“can be interpreted as a kind of natural gradient descent in γ and ascent in δ”_. However, the method referred to simply follows the standard gradient. We asked one of the authors about the meaning behind the statement through private correspondence, to which they replied _“I wouldn't dare to claim that there is a direct connection to Amari's natural gradient, perhaps more to his idea of information geometry and em algorithms.”_. In any case, the statement is about the gradient-based method and not the “standard” updates. We are not aware of any other work by Heskes suggesting a connection between EP and NGD.
>
> 7. Wilkinson et al. (2021) present a unifying framework encompassing several algorithms, viewing them as performing online Newton optimisations with localised updates. This framework elegantly illustrates connections between EP, VI and posterior linearisation. Their view is complementary to ours, and does not make any claims about a connection between EP and NGD in the general case. The authors do however show that the updates of power EP coincide with those of natural gradient VI when the variational limit of the power parameter is taken ($\beta_i \rightarrow \infty$, in our notation); this point was also made by Bui et al. (2018). Our connection is more general, and applies for any value of $\beta_i$. This result is clearly relevant however, and we will reference it after Proposition 1 on line 127.

---

> > ### Comment · Reviewer_6XAG · 2024-08-12
> >
> > Thank you for your response. I think further clarifying and being more specific with the distinction with SNEP (Hasenclever et al. [13]) would be beneficial to the paper. For example on line 219 you state that SNEP can be viewed as doing natural gradient descent 'but with distributions that are more closely matched with those being optimised'  however it is not clear to me what 'distributions that are more closely matched with those being optimised'.
> >
> > In light of your response and the other reviews, I will increase my score by one point.

---

### Author Rebuttal · Authors · 2024-08-07

We would like to thank all of the reviewers for taking the time to review our paper. All reviewers offered valuable feedback, which we have taken on board. Two reviewers highlighted weaknesses related to the paper presentation; we have taken specific steps to address the points raised, which are detailed in the rebuttals below. One reviewer questioned the novelty of our NGD interpretation of EP; we believe we have addressed this below, and will add wording to the paper to make the extent of our contribution clearer. Two reviewers also asked for variational inference to be included as an additional baseline, which we have committed to doing for the camera-ready paper.

---

### Decision · Program_Chairs · 2024-09-25

**Decision:**

Accept (spotlight)

**Comment:**

This paper presents a new view of the moment-matching updates of expectation propagation (EP) as  natural gradient descent (NGD) of a variational objective. This view allows the authors to propose two new methods that address the deficiencies of previous approaches, in particular that of not being robust to Monte Carlo noise when estimating the corresponding expectations.

Overall, the reviews acknowledge the novelty of the approach (which has been further clarified in the rebuttal) and have praised the paper for the “clever” trick of reducing the bias of the MC expectation via change of parameterization and for being technically very strong. Indeed, one of the reviewers mentions “I'm indeed impressed that EP can work with single-sample estimation in the inner loop”. The main issues raised by the reviewers concern (1) the clarity of the presentation of the EP algorithm, (2) description of the results and (3) relation to previous work (in particular wrt to the work of Hasenclever et al, 2017). I believe the authors have addressed these issues in their rebuttal successfully with some of the reviewers updating their score accordingly.

With this, I recommend acceptance.